# Conf-Gen: Conformal Uncertainty Quantification for Generative Models

**Gabriel Loaiza-Ganem** [1]  **Kevin Zhang** [1]  **Wei Cui** [1]  **Marc T. Law** [1]  **Kin Kwan Leung** [1]

## Abstract

Conformal prediction (CP) and its extension, conformal risk control (CRC), are established frameworks for quantifying uncertainty in supervised machine learning through formal guarantees. However, recent breakthroughs in artificial intelligence (AI) have been driven by unsupervised generative models, such as large language models (LLMs) and image generators, which are not directly compatible with CP or CRC. In this work we introduce *conformal generation* (Conf-Gen), a general framework adapting CRC to generative tasks while relaxing its theoretical assumptions. Conf-Gen unifies and generalizes previous attempts to apply CP to LLMs, and extends conformal methodology to entirely new domains. We demonstrate the flexibility of Conf-Gen through some novel applications, including obtaining conformal guarantees on: image generators producing non-memorized images, conversational AI systems having asked enough clarifying questions, and the output of AI agents being correct.

## 1. Introduction

Significant empirical breakthroughs have been achieved in artificial intelligence (AI) in recent years, from the advent of large language models (LLMs; Brown et al., 2020; OpenAI, 2024), to realistic image generators (Ramesh et al., 2022; Rombach et al., 2022), agentic systems (Park et al., 2023), and more (Silver et al., 2016; Potapenko et al., 2021). Most of these developments have moved away from supervised learning and instead leverage generative models. Despite these strides, uncertainty quantification (UQ) remains a critical challenge in AI, particularly for generative models, thereby hindering their deployment in high-stakes applications such as healthcare (Begoli et al., 2019) and scientific discovery (Wang et al., 2023a).

Conformal prediction (CP; Gammerman et al., 1998; Vovk et al., 1999; Saunders et al., 1999; Gammerman & Vovk, 2007) is a principled approach for UQ in supervised learning. CP quantifies uncertainty by using features $X$ to construct a set $\mathcal{C}(X)$ of predictions guaranteed to contain the ground truth $Y_{\text{GT}}$ with a user-specified probability. Conformal risk control (CRC; Angelopoulos et al., 2024b) extends CP by lower-bounding the expectation of a more general utility function, $U(\mathcal{C}(X), Y_{\text{GT}})$, of which CP is a special case. Because conformal guarantees are distribution-free, they are particularly appealing as an approach for UQ capable of handling the complex distributions represented by generative models. Yet, despite its generality, CRC is not directly applicable to generative tasks. As a consequence, work leveraging conformal ideas in the context of generative models tends to be highly task-specific.

In this work we make the following contributions:

(a) **Framework.** We propose *conformal generation* (Conf-Gen, depicted in Figure 1), a framework extending CRC for rigorous UQ in generative modelling. Conf-Gen can be applied to structures beyond sets, such as sequences, and formally lower bounds the expected admissibility (the analogue of $U$ for generative tasks) of the selected structures. We show that existing task-specific conformal methods can be recovered as special cases of Conf-Gen.

(b) **Theory.** Conf-Gen relaxes the theoretical assumptions of CRC. Most importantly, we weaken a monotonicity assumption and we derive a more general reverse (upper) bound on the expected utility/admissibility.

(c) **Python Package.** Naïvely carrying out the computations required by Conf-Gen is sometimes intractable. We identify patterns in the design space of Conf-Gen components which enable efficient computations, and provide a flexible Python package (`https://github.com/layer6ai-labs/conf-gen`) supporting the full range of Conf-Gen applications.

(d) **Empirical Validation.** We demonstrate that Conf-Gen outperforms state-of-the-art conformal baselines in LLM question answering and showcase its versatility across various novel tasks. Specifically, we show conformal guarantees for: generating non-memorized images; ensuring a conversational LLM has asked enough

[1]Layer 6 AI, Toronto, Canada. Correspondence to: All authors <{gabriel, kevinz, wei, marc, kk}@layer6.ai>.

*Proceedings of the 43rd International Conference on Machine Learning*, Seoul, South Korea. PMLR 306, 2026. Copyright 2026 by the author(s).

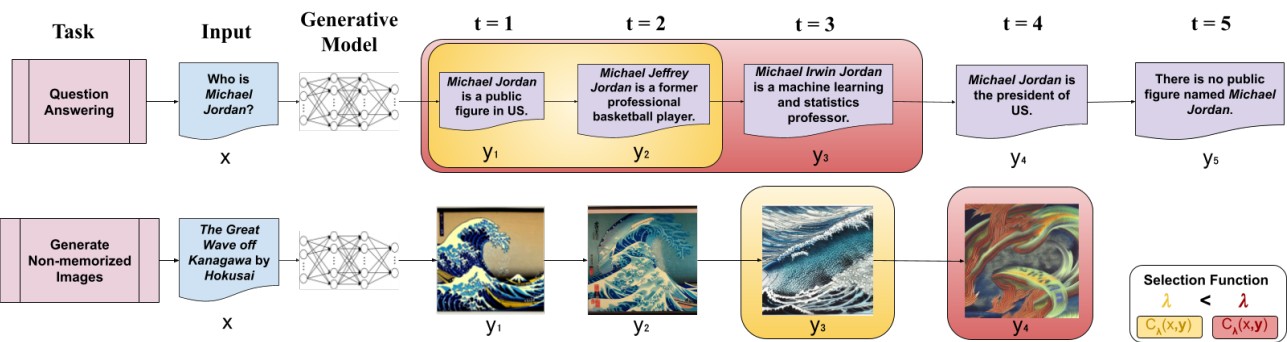

*Figure 1.* Simplified illustration of our method. The selection function $\mathbf{C}_\lambda(x, \mathbf{y})$ processes an input $x$ and a sequence of generations $\mathbf{y}$ to yield an output that becomes increasingly conservative as $\lambda$ increases. Conf-Gen identifies a value $\hat{\lambda}$ that formally ensures the output of $\mathbf{C}_{\hat{\lambda}}$ satisfies a target *admissibility* requirement in expectation. **Top**: when $\mathbf{y}$ consists of *i.i.d.* responses to a query $x$, admissibility ensures the output contains at least one correct answer. **Bottom**: when the images in $\mathbf{y}$ are generated by progressively modifying the prompt $x$, admissibility ensures the outputted image is not memorized.

clarifying questions; producing a sequence of agentic AI outputs containing a correct solution; and selecting a subset of trees from a random forest (Breiman, 2001) containing enough trees making the correct prediction.

## 2. Conformal Prediction and Risk Control

**Split Conformal Prediction.** We present a short summary of key concepts from CP that are relevant to Conf-Gen; for more thorough presentations, we refer the reader to Vovk et al. (2005), Angelopoulos & Bates (2021), and Angelopoulos et al. (2024a). We focus on split conformal prediction (Papadopoulos et al., 2002), a particular instance of CP; henceforth, we will use the abbreviation CP to refer to split conformal prediction. In CP, we assume access to a calibration dataset containing features $X^{(i)} \in \mathcal{X}$ and their corresponding ground truth target $Y_{\mathrm{GT}}^{(i)} \in \mathcal{Y}_{\mathrm{GT}}$ for $i = 1, \ldots, n$. Then, given a new $X^{(n+1)}$, the goal of CP is to use the calibration dataset to produce a set $\mathcal{C}(X^{(n+1)}) \subseteq \mathcal{Y}_{\mathrm{GT}}$ which contains $Y_{\mathrm{GT}}^{(n+1)}$ with probability at least $\gamma$, where $\gamma$ is a user-specified quantity. To achieve this goal, CP uses a score function $S^\downarrow : \mathcal{X} \times \mathcal{Y}_{\mathrm{GT}} \to \mathbb{R}$, with large values of $S^\downarrow(x, y)$ conveying low confidence that $y$ is compatible with $x$. For example, in a classification setting, we could have $S^\downarrow(x, y) = 1 - S^\uparrow(x, y)$, where $S^\uparrow(x, y)$ is the probability assigned by a pre-trained classifier to $y$ when given $x$ as input. For a given $\lambda \in \mathbb{R}$, CP considers sets of the form:

$$\mathcal{C}_\lambda(x) \coloneqq \{y \in \mathcal{Y}_{\mathrm{GT}} : S^\downarrow(x, y) \le \lambda\}. \quad (1)$$

CP performs *calibration* by setting $\hat{\lambda}$ to be the $\frac{\lceil (n+1)\gamma \rceil}{n}$ quantile of $S^\downarrow(X^{(1)}, Y_{\mathrm{GT}}^{(1)}), \ldots, S^\downarrow(X^{(n)}, Y_{\mathrm{GT}}^{(n)})$. Then, $\mathcal{C}_{\hat{\lambda}}(X^{(n+1)})$ satisfies the CP desiderata:

**Theorem 1** (Vovk et al. (1999)). *Assume* $(X^{(1)}, Y_{\mathrm{GT}}^{(1)}), \ldots, (X^{(n+1)}, Y_{\mathrm{GT}}^{(n+1)})$ *are exchangeable. Then, for* $\mathcal{C}_\lambda$ *and* $\hat{\lambda}$ *as defined above:*

$$\mathbb{P}\left(Y_{\mathrm{GT}}^{(n+1)} \in \mathcal{C}_{\hat{\lambda}}\left(X^{(n+1)}\right)\right) \ge \gamma. \quad (2)$$

Recall that the random variables $Z^{(1)}, \ldots, Z^{(n+1)}$ are *exchangeable* if their joint distribution is invariant under any permutation of their indices. Exchangeability is a remarkably flexible assumption, as it is strictly weaker than the *i.i.d.* requirement. Two remarks are worth making about Theorem 1. First, note that the probability in Equation 2 is taken with respect to all random quantities inside it, including the calibration dataset itself (upon which $\hat{\lambda}$ depends). Second, although the conformal guarantee in Equation 2 holds regardless of the underlying data-generating distribution and the choice of $S^\downarrow$, the practical usefulness of CP does depend on $S^\downarrow$; for example, a poor choice of $S^\downarrow$ could lead to $\mathcal{C}_{\hat{\lambda}}(X^{(n+1)}) = \mathcal{Y}_{\mathrm{GT}}$, which would render the conformal guarantee vacuously true.

**Conformal Risk Control (CRC).** Angelopoulos et al. (2024b) extended CP to a more general setting where rather than controlling for coverage, the expectation of an arbitrary utility function is controlled instead.[1] Let $\mathcal{C}_\lambda : \mathcal{X} \to 2^{\mathcal{Y}_{\mathrm{GT}}}$ be a family of functions, indexed by $\lambda \in \Lambda \coloneqq [\lambda_{\min}, \lambda_{\max}]$, producing prediction sets, and let $U : 2^{\mathcal{Y}_{\mathrm{GT}}} \times \mathcal{Y}_{\mathrm{GT}} \to [0, \infty]$ be a utility function on prediction sets. Intuitively, larger values of $\lambda$ should result in more conservative prediction sets, i.e., ones achieving higher utility. CRC performs calibration by finding the smallest $\lambda$ such that the average utility on the calibration set exceeds $\frac{n+1}{n}\gamma$. Formally,

$$\hat{\lambda} = \inf\left\{\lambda \in \Lambda : \bar{U}_n(\lambda) \ge \frac{n+1}{n}\gamma\right\} \wedge \lambda_{\max}, \quad (3)$$

where $\bar{U}_n(\lambda) \coloneqq \frac{1}{n}\sum_{i=1}^n U^{(i)}(\lambda)$ and $U^{(i)}(\lambda) \coloneqq U(\mathcal{C}_\lambda(X^{(i)}), Y_{\mathrm{GT}}^{(i)})$.[2] Under some regularity conditions, a conformal guarantee follows:

---

[1]CRC is usually formulated as upper-bounding a loss function. Lower-bounding a utility function is equivalent, and will facilitate comparisons to our method.

[2]Recall that $\wedge$ and $\vee$ denote the minimum and maximum between two numbers, respectively, and that $\inf \emptyset = \infty$; the minimum in Equation 3 handles this case.

**Theorem 2** (Angelopoulos et al. (2024b))**.** *Assume the functions $U^{(1)}, \ldots, U^{(n+1)}$ are exchangeable, and that they are right-continuous and non-decreasing on $\Lambda$, almost surely. Assume also that $U^{(i)}(\lambda_{\max}) \geq \gamma$, almost surely. Then, for $\hat{\lambda}$ defined as in Equation 3, the following inequality holds:*

$$\mathbb{E}\left[U^{(n+1)}(\hat{\lambda})\right] \geq \gamma. \tag{4}$$

Note that the functions $U^{(1)}, \ldots, U^{(n+1)}$ are exchangeable whenever $(X^{(1)}, Y_{\mathrm{GT}}^{(1)}), \ldots, (X^{(n+1)}, Y_{\mathrm{GT}}^{(n+1)})$ are exchangeable, provided $U$ and $\mathcal{C}_\lambda$ are either deterministic, or independent of the features and ground truth variables. Once again, the expectation in Equation 4 is taken with respect to all random quantities inside it.

The lower bounds in Theorem 1 and Theorem 2 represent the foundational guarantees of CP and CRC, respectively. Corresponding upper bounds quantifying the degree of overcoverage in the resulting sets also exist; we summarize them in Appendix A. Note that CRC generalizes CP; indeed, Equation 4 recovers Equation 2 when $U(\mathcal{C}, y) = \mathbb{1}(y \in \mathcal{C})$.

## 3. Conformal Generation

**Motivation.** The goal of this section is to extend conformal methodology to generative tasks in a highly general way. We begin by identifying several practical limitations in the current application of CRC. First, while Theorem 2 does not explicitly restrict $\mathcal{X}$ to the feature space of a supervised learning problem, Angelopoulos et al. (2024b) focus exclusively on this setting. Second, $\mathcal{C}_\lambda$ always outputs sets, despite this not being a prerequisite to maintain exchangeability. Third, current implementations assume $U$ is a callable function, which precludes utilities defined by procedures such as human evaluation. Finally, the requirement that $U^{(i)}$ be non-decreasing is overly restrictive for some generative tasks. We will see that Conf-Gen addresses all these limitations.

**Notation.** We use calligraphic font to denote sets (e.g., $\mathcal{X}$ and $\mathcal{Y}$), and we write the space of finite $\mathcal{Y}$-valued sequences as $\mathcal{Y}^*$, i.e., $\mathcal{Y}^* := \cup_{t=0}^\infty \mathcal{Y}^t$. We use lower-case letters to denote fixed elements of these sets (e.g., $x \in \mathcal{X}$ and $y \in \mathcal{Y}$), upper-case letters to denote random variables (e.g., $X \in \mathcal{X}$ and $Y \in \mathcal{Y}$), and bold font to denote tuples and sequences (e.g., a random sequence $\mathbf{Y} = (Y_1, \ldots, Y_T) \in \mathcal{Y}^*$). We denote the length of a sequence $\mathbf{y}$ as $|\mathbf{y}|$, and its subsequence given by $(y_1, \ldots, y_t)$ as $\mathbf{y}_{:t}$.

**Setting.** In Conf-Gen we observe samples $\mathbf{G} = (X, \mathbf{Y}, Y_{\mathrm{GT}})$ from some distribution $\mathbb{P}_{\mathbf{G}}$ on $\mathcal{X} \times \mathcal{Y}^* \times \mathcal{Y}_{\mathrm{GT}}$. Here, $X$ corresponds to a conditioning variable or "input" of the generative model, $\mathbf{Y}$ is a sequence of generations or "outputs" from the model, and $Y_{\mathrm{GT}}$ is an optional ground truth corresponding to $X$. For example, $\mathcal{X}$ and $\mathcal{Y}$ could both

be the space of text, with $X$ corresponding to a question, $\mathbf{Y}$ to a sequence of answers to $X$ generated by an LLM, and $Y_{\mathrm{GT}}$ to a ground truth answer; we will use this as a running example throughout our paper, but we nonetheless highlight that our method is much more general. The goal of Conf-Gen will be to use $(X, \mathbf{Y})$, which plays the role of the "features" in CRC, to produce a set (or more generally, a structured output, such as a sequence) guaranteed to have large expected admissibility, which expands the notion of utility in CRC to the generative realm. Conf-Gen provides partial control over $\mathbb{P}_{\mathbf{G}}$, whereas CRC lacks control over its data-generating distribution. For instance, in our running example, we can choose how each individual answer $Y_t$ in $\mathbf{Y}$ is generated (e.g., they could be generated independently, or we could also enforce the LLM to provide a different answer than its previous ones, $Y_1, \ldots, Y_{t-1}$).

The Conf-Gen framework is comprised by three components: a parameterized family of selection functions, an admissibility function, and a calibration dataset. We define and discuss these components below.

**Selection Functions.** We assume access to a family of potentially stochastic functions, $\mathbf{C}_\lambda : \mathcal{X} \times \mathcal{Y}^* \to \mathcal{S}$, indexed by $\lambda \in \Lambda$, where $\Lambda \subseteq [-\infty, \infty]$ is such that $\infty \in \Lambda$. Here, $\mathcal{S}$ is the output space over which we will provide a conformal guarantee; it is precisely the freedom to choose $\mathcal{S}$ that allows Conf-Gen to extend CRC beyond sets. Natural choices of $\mathcal{S}$ include $\mathcal{S} = 2^{\mathcal{Y}}$ and $\mathcal{S} = \mathcal{Y}^*$; most examples in our paper will use the latter, and we hence write $\mathbf{C}_\lambda$ in bold font. Intuitively, larger values of $\lambda$ should correspond to $\mathbf{C}_\lambda$ producing more conservative outputs, i.e., less useful but achieving larger admissibility values. $\mathbf{C}_\lambda$ is analogous to $\mathcal{C}_\lambda$ from CP and CRC, but it takes a sequence of generations as an additional input. In our running example, $\mathbf{C}_\lambda(x, \mathbf{y})$ could be given by the subsequence of $\mathbf{y}$ containing only the elements $y_t$ such that $S^\downarrow(x, y_t) \leq \lambda$, where $S^\downarrow : \mathcal{X} \times \mathcal{Y} \to \mathbb{R}$ is a score function, e.g., $S^\downarrow(x, y) = -S^\uparrow(x, y)$, where $S^\uparrow(x, y)$ is obtained by asking the LLM to provide a scalar value expressing its confidence that $y$ is a correct answer to the question $x$, or by having access to the likelihood function of the LLM.[3]

**Admissibility Function.** We also assume the existence of a potentially stochastic admissibility function $A : \mathcal{X} \times \mathcal{S} \times \mathcal{Y}_{\mathrm{GT}} \to [0, \infty]$, where $A(X, \mathbf{C}_\lambda(X, \mathbf{Y}), Y_{\mathrm{GT}})$ generalizes the role of $U(\mathcal{C}_\lambda(X), Y_{\mathrm{GT}})$ in CRC. Intuitively, larger values of $A$ are more desirable, but the outputs of $\mathbf{C}_\lambda$ must become more conservative to increase $A$. In our running example, $A(x, \mathbf{y}, y_{\mathrm{GT}})$ will be a binary variable indicating whether at

---

[3]We will use both $S^\downarrow$ and $S^\uparrow$ to denote score functions, with the convention that smaller values of $S^\downarrow$ are "better" (i.e., expected to have large instance-level admissibility, see Section 4.1; in our LLM example, this means correctness of the answer), whereas larger values of $S^\uparrow$ are "better".

least one element of $\mathbf{y}$ is a correct answer to $x$ (other choices, e.g., a scalar indicating the quality of the best answer in $\mathbf{y}$, could also be possible). When the ground truth is well-defined, this $A$ could be given by an LLM call checking the semantic equivalence between $y_{GT}$ and each element of $\mathbf{y}$. Importantly though, we do not assume that $A$ is callable. For example, the question $x$ being asked could so open-ended that it has no corresponding ground truth answer $y_{GT}$. In this case, $A$ could instead be given by human evaluation without access to $y_{GT}$; mathematically, this corresponds to $A(x, \mathbf{y}, y_{GT})$ ignoring $y_{GT}$, and thus Conf-Gen does not require a well-defined ground truth.

**Calibration Dataset.** Let us consider a sequence $\mathbf{G}_{:n} := ((X^{(i)}, \mathbf{Y}^{(i)}, Y_{GT}^{(i)}))_{i=1}^{n}$ containing $n$ samples from $\mathbb{P}_{\mathbf{G}}$. The last component for Conf-Gen is a calibration dataset, $\mathbf{D}_{:n}$, which contains the admissibility of every tuple in $\mathbf{G}_{:n}$ for every $\lambda \in \Lambda$, i.e.,

$$\mathbf{D}_{:n} := \left( \left\{ A^{(i)}(\lambda) \right\}_{\lambda \in \Lambda} \right)_{i=1}^{n}, \tag{5}$$

where $A^{(i)}(\lambda) := A(X^{(i)}, \mathbf{C}_\lambda(X^{(i)}, \mathbf{Y}^{(i)}), Y_{GT}^{(i)})$. Note that the generated sequences in $\mathbf{G}_{:n}$ need not have the same length; we make this explicit by writing $\mathbf{Y}^{(i)} = (Y_1^{(i)}, \ldots, Y_{T_i}^{(i)})$. We also highlight that while $A$ need not be callable, we do require access to $\mathbf{D}_{:n}$. In our running LLM example, this can be achieved—even when $A$ is not callable—by evaluating the correctness of every answer in $\mathbf{Y}^{(i)}$ for every $i = 1, \ldots, n$. We will cover this point in more detail and generality in Section 4.1.

**Conformal Generation.** For a given instance of Conf-Gen composed of the above components, the goal is to leverage the calibration dataset to find $\hat{\lambda}$ such that $\mathbf{C}_{\hat{\lambda}}$ produces outputs that achieve large admissibility values. In our running example, this means $\mathbf{C}_{\hat{\lambda}}(X^{(n+1)}, \mathbf{Y}^{(n+1)})$ contains a correct answer with high probability. In analogy to CRC, calibration is performed by finding $\hat{\lambda}$, the smallest $\lambda$ such that the average admissibility on the calibration dataset is at least $\frac{n+1}{n}\gamma$. More formally, for $\gamma \geq 0$, we define

$$\hat{\lambda} := \inf \mathcal{V}\left( \mathbf{D}_{:n}, \frac{n+1}{n}\gamma \right), \tag{6}$$

where

$$\mathcal{V}(\mathbf{D}_{:n}, \gamma') := \left\{ \lambda \in \Lambda : \bar{A}_n(\lambda) \geq \gamma' \right\} \cup \{\infty\} \tag{7}$$

and $\bar{A}_n(\lambda) := \frac{1}{n} \sum_{i=1}^{n} A^{(i)}(\lambda)$. We now introduce the notion of a $\gamma$-*sensible* instance of Conf-Gen, which provides the requirements for our conformal guarantees.

**Definition 1.** *We say that an instance of Conf-Gen is $\gamma$-sensible if the following properties hold:*
*(a) The functions $A^{(1)}, \ldots, A^{(n+1)}$ are exchangeable.*

*(b) For every $\gamma', \gamma'' \geq 0$, if $\lambda' = \inf \mathcal{V}(\mathbf{D}_{:n}, \gamma')$ and $\lambda'' = \inf \mathcal{V}(\mathbf{D}_{:n+1}, \gamma'')$, then:*
   *(b1) The map given by $\lambda \mapsto \mathbb{E}[A^{(n+1)}(\lambda) \mid \lambda', \lambda'']$ is non-decreasing on $\Lambda$, almost surely.*
   *(b2) $\lambda' \in \mathcal{V}(\mathbf{D}_{:n}, \gamma')$ and $\lambda'' \in \mathcal{V}(\mathbf{D}_{:n+1}, \gamma'')$, almost surely.*
*(c) $A(X, \mathbf{C}_\infty(X, \mathbf{Y}), Y_{GT}) \geq \gamma$, almost surely.[4]*

Property $(a)$, which is identical to the exchangeability assumption of CRC, holds whenever $\mathbf{G}_{:n+1}$ is exchangeable and both $A$ and $\mathbf{C}_\lambda$ are independent of $\mathbf{G}_{:n+1}$. Note that in our running example this property does not require tokens within an answer, nor answers within a sequence of answers, to be exchangeable; we only require different sequences of answers themselves to be exchangeable. Intuitively, property $(b1)$ weakens the strict monotonicity required by CRC to "sufficient monotonicity", and $(b2)$ relaxes the right-continuity assumption of CRC. Lastly, $(c)$ is analogous to the condition that $U(\mathcal{C}_{\lambda_{max}}(X), Y_{GT}) \geq \gamma$ in CRC, although we make it explicit that $\Lambda$ need not be bounded by defaulting its maximal element to $\infty$ and requiring that $\infty \in \Lambda$.

**Theorem 3.** *Consider a $\gamma$-sensible instance of Conf-Gen. Then, for $\hat{\lambda}$ as defined in Equation 6,*

$$\mathbb{E}\left[ A^{(n+1)}(\hat{\lambda}) \right] \geq \gamma. \tag{8}$$

*Additionally, if $A : \mathcal{X} \times \mathcal{S} \times \mathcal{Y}_{GT} \to [0, a_{max}]$, then*

$$\mathbb{E}\left[ A^{(n+1)}(\hat{\lambda}) \right] \leq \gamma + \frac{a_{max}}{n+1} + \mathbb{E}[H], \tag{9}$$

*where $\lambda^{**} := \inf \mathcal{V}(\mathbf{D}_{:n+1}, \gamma + \frac{a_{max}}{n+1})$ and*

$$H := \bar{A}_{n+1}(\lambda^{**}) - \sup\left\{ \bar{A}_{n+1}(\lambda) : \lambda \in \Lambda, \lambda < \lambda^{**} \right\} \vee 0. \tag{10}$$

*Proof.* See Appendix B. □

**Conf-Gen vs CRC.** Despite Definition 1 being relatively technical, Conf-Gen relies on strictly weaker assumptions than CRC. Most notably, it relaxes the monotonicity assumption from an almost sure condition to one in conditional expectation. It also enables the construction of conformal objects beyond sets ($\mathcal{S} = 2^{\mathcal{Y}}$), such as sequences ($\mathcal{S} = \mathcal{Y}^*$), along with other minor technical refinements. Conf-Gen serves as a strict generalization of CRC, where any instance of the latter can be recovered as a specific case of the former (by having $\mathbf{C}_\lambda$ ignore $\mathbf{Y}$, and $A$ ignore $X$). The gap between the lower bound in Equation 8 and the upper bound

---

[4]In practice, this property can always be achieved. For example, we can add an element, Abstain, to $\mathcal{Y}$, define $A(x, \mathbf{y}, y_{GT})$ to be $\gamma$ (or any larger scalar) whenever Abstain is an element of $\mathbf{y}$, and define $\mathbf{C}_\infty(x, \mathbf{y})$ to always have Abstain as an element. Intuitively, a sequence containing Abstain represents that we abstain from making a non-vacuous conformal guarantee, analogously to having $\mathcal{C}_{\hat{\lambda}}(X^{(n+1)}) = \mathcal{Y}_{GT}$ in CP.

in Equation 9 is determined by $\mathbb{E}[H]$, where, intuitively, $H$ quantifies the size of the jumps of the average admissibility, $\bar{A}_{n+1}$. Importantly, Equation 9 generalizes the existing upper bound for CRC. A detailed discussion of the theoretical assumptions underpinning Conf-Gen, as well as a comprehensive comparison to CRC, is provided in Appendix C.

## 4. Computational Considerations

In this section we outline common patterns in the selection and admissibility functions which enable efficient calibration and inference within Conf-Gen. Our accompanying Python package supports all the instances and patterns we mention, as well as the straightforward customization of all Conf-Gen components. Beyond the examples presented here, our package includes a broader suite of utilities designed to support a wide range of Conf-Gen applications.

### 4.1. Efficient Calibration

In practice, the procedure for finding $\hat{\lambda}$ must be tailored to the particular Conf-Gen instance at hand. If $\bar{A}_n$ is suitable to gradient-based optimization, $\hat{\lambda}$ can be obtained by calling a solver to minimize $\lambda$ subject to $\bar{A}_n(\lambda) \geq \frac{n+1}{n}\gamma$. If $\Lambda$ is finite, constructing $\mathbf{D}_{:n}$ from $\mathbf{G}_{:n}$ requires evaluating the admissibility function $n \times |\Lambda|$ times, after which computing $\hat{\lambda}$ is straightforward; this process could range from trivially cheap to prohibitively expensive, depending on the cost of evaluating $A$. If $\Lambda$ is infinite and $A$ is not a callable function (e.g., if it is given by human evaluation), calibration might seem out of reach since calling a solver is not feasible and constructing $\mathbf{D}_{:n}$ appears impossible.

**Score-Based Selection Functions.** Specific choices of $\mathbf{C}_\lambda$ enable calibration even in this challenging scenario. Our key insight here is that if $\mathbf{C}_\lambda$ has a finite image as a function of $\lambda$ (i.e., if $\{\mathbf{C}_\lambda(X, \mathbf{Y}) : \lambda \in \Lambda\}$ is finite, almost surely), then we can characterize $\mathbf{D}_{:n}$—even if the set $\Lambda$ is infinite— by evaluating $A^{(i)}$ at the finitely-many possible outputs of $\mathbf{C}_\lambda(X^{(i)}, \mathbf{Y}^{(i)})$, for every $i = 1, \ldots, n$. A finite image can be achieved by defining $\mathbf{C}_\lambda(x, \mathbf{y})$ as a subsequence of $\mathbf{y}$ derived from scores assigned to each element in $\mathbf{y}$. In the example from Section 3, the selection function uses scores $S_t^\downarrow = S^\downarrow(x, y_t)$ to form the subsequence $\mathbf{C}_\lambda(x, \mathbf{y}) = (y_t : S_t^\downarrow \leq \lambda)$. Other constructions are also viable; for example, $\mathbf{C}_\lambda(x, \mathbf{y})$ could be defined as $\mathbf{y}_{:\tau(x,\mathbf{y},\lambda)}$, where $\tau(x, \mathbf{y}, \lambda)$ is the first index $t$ at which the accumulated score of $\mathbf{y}_{:t}$ exceeds $\lambda$. More formally,

$$\tau(x, \mathbf{y}, \lambda) \coloneqq \inf\left\{t : \texttt{accum}\left(S_1^\uparrow, \ldots, S_t^\uparrow\right) > \lambda\right\} \wedge |\mathbf{y}|, \quad (11)$$

where $S_t^\uparrow = S^\uparrow(x, y_t)$ and $\texttt{accum} : \mathbb{R}^* \to \mathbb{R}$ represents an accumulation function such as a sum or a maximum. A sum would be appropriate when the individual scores are non-negative, and in the context of our running example,

if the individual answers tend to be varied. Conversely, a maximum might be preferable if duplicate answers are frequent, as it prevents redundant answers from inflating the accumulated score. These examples are illustrative rather than exhaustive, for instance: the rules for generating a subsequence from individual scores are flexible, the score at time $t$ could depend on the entire prefix $\mathbf{y}_{:t}$ rather than just the individual element $y_t$, and the finite image property can still be satisfied when $\mathbf{C}_\lambda(x, \mathbf{y})$ is not a subsequence of $\mathbf{y}$.

**Admissibility Evaluation.** Without additional structure beyond a finite image, calibration requires evaluating $A$ across all potential outputs of $\mathbf{C}_\lambda$; this can be suboptimal because the evaluation of $A$ becomes tied to the specific choice of $\mathbf{C}_\lambda$. For example, testing a new selection function would require a completely new set of evaluations $\mathbf{D}_{:n}$, even if $\mathbf{G}_{:n}$ remains fixed. To address this issue, we can often identify an instance-level admissibility function, $A' : \mathcal{X} \times \mathcal{Y} \times \mathcal{Y}_{\text{GT}} \to [0, \infty]$, which allows $A$ to be decomposed as

$$A(x, \mathbf{y}, y_{\text{GT}}) = \texttt{agg}(A_1', \ldots, A_T'), \quad (12)$$

where $A_t' \coloneqq A'(x, y_t, y_{\text{GT}})$ and $\texttt{agg} : \mathbb{R}^* \to \mathbb{R}$ is an aggregation function, such as the minimum or maximum. In our running example, $\texttt{agg}$ is the maximum and $A_t'$ represents the correctness of $y_t$ as an answer to $x$. Under this decomposition, when $\mathbf{C}_\lambda(x, \mathbf{y})$ is a subsequence of $\mathbf{y}$, evaluating $A'(X^{(i)}, Y_t^{(i)}, Y_{\text{GT}}^{(i)})$ for all $t = 1, \ldots, T_i$ and $i = 1, \ldots, n$ is sufficient for calibration. This process decouples the evaluation of $A$ from the choice of $\mathbf{C}_\lambda$, as it requires evaluating $A'$ over the same fixed set of $\sum_{i=1}^n T_i$ points, regardless of $\mathbf{C}_\lambda$. Lastly, as with selection functions, this example is not exhaustive, e.g., $A_t'$ could depend on the prefix $\mathbf{y}_{:t}$ rather than just the single element $y_t$.

### 4.2. Efficient Inference through Partial Generation

A priori, once calibration has been performed, computing $\mathbf{C}_{\hat{\lambda}}(X^{(n+1)}, \mathbf{Y}^{(n+1)})$ at test time requires access to both $X^{(n+1)}$ and $\mathbf{Y}^{(n+1)}$; however, many scenarios only necessitate partial access to these quantities. We refer to these cases as having the capability for *partial generation*. For instance, there may exist a stopping time $\tau_{n+1} \leq T_{n+1}$ such that $\mathbf{Y}_{:\tau_{n+1}}^{(n+1)}$ contains all the information needed from $\mathbf{Y}^{(n+1)}$ to compute $\mathbf{C}_{\hat{\lambda}}(X^{(n+1)}, \mathbf{Y}^{(n+1)})$. When this is the case, we can sequentially generate $Y_1^{(n+1)}, \ldots, Y_{\tau_{n+1}}^{(n+1)}$ and stop, thus avoiding having to generate the remaining $T_{n+1} - \tau_{n+1}$ elements; this can save significant compute when generation is expensive. In the case where $\mathbf{C}_\lambda(x, \mathbf{y}) = \mathbf{y}_{:\tau(x,\mathbf{y},\lambda)}$ for $\tau(x, \mathbf{y}, \lambda)$ defined as in Equation 11, we have that $\tau_{n+1} = \tau(X^{(n+1)}, \mathbf{Y}^{(n+1)}, \hat{\lambda})$. Notably, the potential for partial generation extends beyond this single instance.

### 4.3. Examples

Not all design choices within Conf-Gen are mutually compatible, i.e., they do not always result in $\gamma$-sensibility holding. Table 1 presents various score-based selection functions alongside candidate admissibility functions with which they are compatible. These entries, which also detail partial generation capabilities, assume $\Lambda = [-\infty, \infty]$ and the instance-level decomposition defined in Equation 12. Further examples are provided in Appendix D.

The first two rows of Table 1 are suited for scenarios where the goal is to ensure that at least one element in $\mathbf{C}_\lambda$ possesses high admissibility (agg = max). In the second row, the nesting property, $\mathbf{C}_\lambda \subseteq \mathbf{C}_{\lambda'}$ for $\lambda \leq \lambda'$, holds inherently. For the first row, this nesting is satisfied provided that accum = max or the score function is non-negative, as either condition ensures that $\tau(x, \mathbf{y}, \lambda)$ is non-decreasing in $\lambda$. In both cases, this nesting behavior ensures that the output of $\mathbf{C}_\lambda$ becomes more conservative as $\lambda$ increases: since appending elements to a sequence cannot decrease the admissibility of its most admissible member, the mapping $\lambda \mapsto A(\lambda) \coloneqq A(X, \mathbf{C}_\lambda(X, \mathbf{Y}), Y_{\text{GT}})$ is non-decreasing, and thus property $(b1)$ of the $\gamma$-sensibility requirements (Definition 1) is satisfied.

The third row is appropriate when the goal is for all elements in $\mathbf{C}_\lambda$ to have large admissibility (agg = min). Here, the nesting property is reversed: $\mathbf{C}_{\lambda'} \subseteq \mathbf{C}_\lambda$ whenever $\lambda \leq \lambda'$. Nevertheless, the outputs of $\mathbf{C}_\lambda$ still become more conservative as $\lambda$ increases because removing elements from a set cannot decrease the admissibility of its least admissible member; consequently, $\lambda \mapsto A(\lambda)$ remains non-decreasing. The conformal factuality framework (Mohri & Hashimoto, 2024), discussed further in Section 5, constitutes an instance of this configuration.

In the fourth row, $\mathbf{C}_\lambda$ always outputs a single element ($\mathcal{S} = \mathcal{Y}$); this is appropriate when the goal is for this element to have high admissibility (the aggregation function becomes irrelevant as long as agg$(y) = y$). Here, since nesting does not hold, $\mathbf{C}_\lambda$ becoming increasingly conservative as $\lambda$ grows becomes contingent on $\mathbb{P}_\mathbf{G}$; thus, formally verifying property $(b1)$ from Definition 1 is harder. Regardless, $\gamma$-sensibility should still hold provided that $\tau(x, \mathbf{y}, \lambda)$ is non-decreasing in $\lambda$ and that $\mathbb{P}_\mathbf{G}$ was chosen appropriately. For example, if $\mathbf{Y}$ consists of responses generated by progressively more capable LLMs, $\mathbf{C}_\lambda$ would return the first response believed to be correct. While a stronger LLM might occasionally produce a worse answer than a weaker one—preventing $\lambda \mapsto A(\lambda)$ from being almost surely non-decreasing—the required monotonicity may still hold in conditional expectation.

Notably, all examples presented here are specifically constructed to ensure the right-continuity of the mapping

*Table 1.* Representative configurations of Conf-Gen.

| Selection function | | Compatible with | |
|---|---|---|---|
| $\mathbf{C}_\lambda(x, \mathbf{y})$ | accum | agg | Partial generation |
| $\mathbf{y}_{:\tau(x,\mathbf{y},\lambda)}$ | max/sum | max | Yes |
| $\{y_t \mid S_t^{\downarrow} \leq \lambda\}$ | N/A | max | No |
| $\{y_t \mid S_t^{\uparrow} > \lambda\}$ | N/A | min | No |
| $y_{\tau(x,\mathbf{y},\lambda)}$ | max | max/min | Yes |

$\lambda \mapsto A(\lambda)$, thus satisfying property $(b2)$ of Definition 1. This property is delicate; for instance, replacing strict inequalities with non-strict ones (or vice versa) within our definitions would generally violate this requirement.

## 5. Related Work as Instances of Conf-Gen

UQ remains a more open problem for generative models than for supervised learning, with current research largely concentrated on LLMs. One prominent direction involves *verbalized uncertainty*, where the model is prompted to provide a scalar estimate of its own confidence (Kadavath et al., 2022; Yang et al., 2024b). Another category is based on having the LLM generate multiple answers and aggregating the variability of these answers into a scalar measure of uncertainty (Wang et al., 2023b; Kuhn et al., 2023; Lin et al., 2024; Grewal et al., 2024; Hou et al., 2024; Yang et al., 2024a; Gao et al., 2024; Qiu & Miikkulainen, 2024; Nikitin et al., 2024; Wang & Holmes, 2025; Ross et al., 2026). These methods provide no formal guarantees and are orthogonal to Conf-Gen; indeed, they can in principle be used to define score functions within Conf-Gen.

Several works have applied conformal ideas to generative models. Teneggi et al. (2023) apply CRC to image-to-image regression tasks with diffusion models (Sohl-Dickstein et al., 2015; Ho et al., 2020; Song et al., 2021), and Quach et al. (2024) and Kladny et al. (2025) apply it to LLMs as in our running example from the previous section, i.e., they aim to produce a set of answers to a question $X$ such that with probability at least $\gamma$, the set contains at least one correct answer. We highlight a few key differences between these two works and ours. First, their admissibility function does not depend on $X$. They thus always assume access to a ground truth answer, and their admissibility function is limited to asking an LLM whether the generated answers match the ground truth. Second, these works sequentially filter generated sets of answers, e.g., to ensure that each answer has high likelihood and that no two answers in the set are too similar (as measured by distance on some embedding space). This choice effectively parameterizes $\mathbf{C}_\lambda$ with multiple values, one per filter, rather than a single scalar $\lambda$. Because each parameter must be calibrated independently on a disjoint subset of the data in an overly conservative way, the procedure ultimately compounds into the production of unnecessarily large conformal sets. Although each of these

fragmented steps are themselves instances of Conf-Gen, our experiments in Section 6 show that an instance of Conf-Gen sidestepping this partitioning yields smaller calibrated sets.

Mohri & Hashimoto (2024) proposed *conformal factuality*, where an LLM answer $Y'$ to a question $X$ is decomposed into a sequence of claims $\mathbf{Y} = (Y_1, \ldots, Y_T)$ via an auxiliary LLM call. Their algorithm produces a subsequence $\mathbf{C}_{\hat{\lambda}}(X, \mathbf{Y})$ where all selected claims are factually correct with probability at least $\gamma$. Kuwahara et al. (2025) adapted this framework to *conformal summarization*, where $\mathbf{Y}$ is now a sequence of sentences making up a document, and their goal is to ensure that some fraction $\beta$ of all important sentences in $\mathbf{Y}$ is retained with probability at least $\gamma$. As noted in Section 4.3, conformal factuality is an instance of Conf-Gen. Conformal summarization can be similarly recovered as an instance of Conf-Gen where the admissibility function is defined as an indicator that the fraction of recovered admissible elements meets the threshold $\beta$ (see Appendix D). We also highlight that both methodologies depend on a nesting property that corresponds to the monotonicity requirement in CRC, which is relaxed in Conf-Gen.

Lastly, in *conformal agent error attribution* (Feng et al., 2026), $X$ is a task for an AI agent, $\mathbf{Y}$ corresponds to a sequence of actions taken by the agent which failed to complete task $X$, and $Y_{\text{GT}}$ indicates the step of the trajectory on which the agent made its first mistake. Feng et al. (2026) show how to obtain a contiguous subsequence of $\mathbf{Y}$ guaranteed to contain the first mistake with probability at least $\gamma$, thus allowing to rollback the agent to the time of its first mistake: this is another instance of Conf-Gen, where a sequence is admissible if it contains the first mistake.

# 6. Experiments

We validate our approach on a diverse set of tasks. Additional details for each experiment, including the precise formulation of every task as an instance of Conf-Gen, are provided in Appendix E. Except for the experiments in Section 6.1, all tasks considered here are novel, representing the first instance of conformal guarantees being provided for these settings. These tasks include both binary and scalar-valued admissibility functions, some of which are not callable. We also consider a case where $\lambda \mapsto A(\lambda)$ is not non-decreasing almost surely, but where $\gamma$-sensibility is expected to hold in practice nonetheless. This variety highlights the wide applicability of Conf-Gen and of our Python package—which we used across all our experiments—as well as the practical relevance of our relaxed theoretical requirements. Code to reproduce all our results is included alongside the package.

All figures in this section contain two panels. Left panels show the average admissibility on a test set as a function of

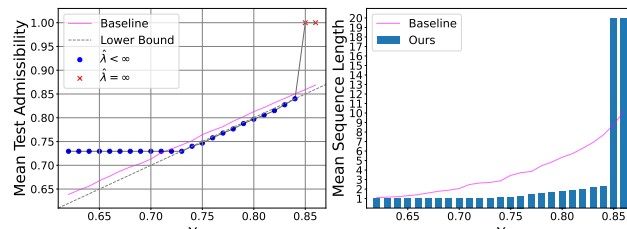

*Figure 2.* Results on test data for the question answering task.

$\gamma$, alongside the corresponding lower bound as a diagonal line. These plots aim to empirically verify the conformal guarantee; we highlight that the few small dips below the diagonal lines do not contradict Theorem 3 as we can only plot empirical averages over a finite test set for a single calibration dataset, not true expectations. Since the conformal guarantee can be trivially achieved by always being maximally conservative, the practical usefulness of Conf-Gen also needs to be empirically verified for each task: this is the goal of the right panels.

## 6.1. Conformal Open-Domain Question Answering

Following our running example, we consider a question answering task (as in the top panel of Figure 1) where responses are generated independently. We utilize the selection function described in the first row of Table 1 (with `accum = max`), except we remove all duplicate entries from $\mathbf{y}_{:\tau(x, \mathbf{y}, \lambda)}$. We benchmark Conf-Gen against the method of Quach et al. (2024) on the TriviaQA dataset (Joshi et al., 2017). For a fair comparison, we adopt their experimental configuration: we use LLaMA-13B (Touvron et al., 2023) as the base LLM and its corresponding length-normalized likelihood (Johnson et al., 2017) as the score function $S^{\uparrow}$. Recall that here, admissibility means that the output contains at least one correct response; consequently, the conformal guarantee ensures that the selected sequence includes such a response with probability at least $\gamma$.

Results are displayed in Figure 2: the left plot confirms that both Conf-Gen and the baseline satisfy the conformal guarantee, while the right plot demonstrates that our method yields more useful outputs by producing shorter sequences across most values of $\gamma$. Additionally, thanks to partial generation, this means that our method also requires fewer LLM calls to produce its output. We provide additional details and experiments for this task in Appendix E.1.

## 6.2. Conformal Generation of Non-Memorized Images

Image generation models such as Stable Diffusion (Rombach et al., 2022) have been shown to memorize some of their training data (Wen et al., 2023). Reproducing memorized images during inference can pose a copyright infringement risk for model providers (Orrick, 2023): this makes

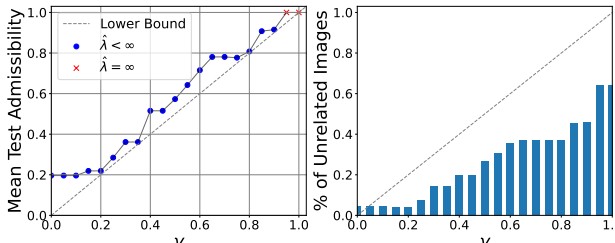

*Figure 3.* Results on test data for generation of non-memorized images.

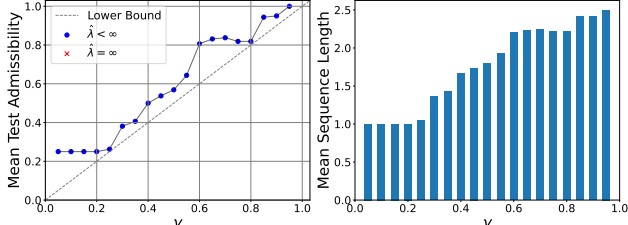

*Figure 4.* Results on test data for the conversational task.

obtaining conformal guarantees on generated images being non-memorized particularly appealing.

In this task, the input is a sequence $\mathbf{X} = (X_1, \ldots, X_T)$, where $X_1$ is a prompt of length $T - 1$ describing an image $Y_{GT}$ that is memorized by Stable Diffusion v1.5, i.e., when given $X_1$, the model produces a near-duplicate of $Y_{GT}$. Subsequent prompts $X_t$ (for $t = 2, \ldots, T$) are generated by an LLM instructed to modify $t - 1$ tokens of $X_1$; we follow the procedure of Ross et al. (2025), which selects tokens deemed to contribute more strongly to memorization first. The sequence $\mathbf{Y}$ contains images generated from these perturbed prompts. We employ the selection function from the final row of Table 1, with the score $S^{\uparrow}$ defined as the negative average norm of the classifier-free guidance term (Ho & Salimans, 2022), which has been shown to be indicative of memorization (Wen et al., 2023).

We rely on the memorized images from Webster (2023) and had their associated generations annotated by human evaluators. Specifically, we had 10 human evaluators assess each generated image and label it as either "good", "medium", or "bad". Images labeled as "good" are those which retain the semantic content of the ground truth image, but did not memorize it; those labeled as "medium", while not memorized, are simply unrelated to the ground truth image; and those labeled as "bad" memorized the ground truth image. The admissibility $A$ is a scalar given by the fraction of 10 human evaluators who classify the generated image as non-memorized relative to the ground truth (i.e., "good" or "medium"). Consequently, the conformal guarantee ensures that, in expectation, the selected image would be classified as non-memorized by at least a fraction $\gamma$ of human evaluators. The bottom panel of Figure 1 provides a simplified illustration of this task.

Note that in this task, $\lambda \mapsto A(\lambda)$ need not be non-decreasing, as an image corresponding to a more heavily modified prompt could, in principle, be deemed more memorized than one corresponding to a less modified prompt. We should of course not expect this to happen often, and thus the $\gamma$-sensibility assumption remains reasonable.

Results are shown in Figure 3. The left panel verifies the conformal guarantee. The right panel displays the average frac-

tion of human evaluators who assessed the selected image as too different from the ground truth (i.e., "medium"). The gap between the left and right plots corresponds to the percentage of "good" images. This gap being large highlights the usefulness of Conf-Gen: it avoids trivially achieving large admissibility by always generating "medium" images. More details are provided in Appendix E.2.

### 6.3. Conformal Conversational AI Chatbot

In this task, the input is also a sequence $\mathbf{X} = (X_1, \ldots, X_T)$, where $X_1$ is now a potentially ambiguous question asked to an AI chatbot. At each step $t$, an LLM generates a candidate answer $Y_t$ to $X_t$; the chatbot then either provides $Y_t$ to the user or requests a clarified version of the prompt, $X_{t+1}$, which incorporates an additional piece of information. We use the selection function given by the last row of Table 1: we interpret the output being $y_{\tau(x,\mathbf{y},\lambda)}$ as meaning that the chatbot answered the question in step $\tau(x, \mathbf{y}, \lambda)$ and requested clarification in all previous steps. The score is obtained by instructing an LLM to provide a value quantifying how ambiguous a question is. The admissibility of a conversation is a binary label equal to 1 if the last question in the conversation is unambiguous. Thus, our conformal guarantee ensures that, with probability at least $\gamma$, the chatbot has asked enough clarifying questions before answering.

We use ClariQ (Aliannejadi et al., 2021), a dataset of questions and clarifications. The results are presented in Figure 4. The left panel verifies the empirical coverage of our conformal guarantee. The right panel illustrates the expected trade-off: as $\gamma$ increases, the chatbot needs to ask more clarifications to ensure admissibility. Importantly, the chatbot does not default to always asking the maximal number of clarifications, indicating the usefulness of Conf-Gen. More details are provided in Appendix E.3.

### 6.4. Conformal Agentic AI for Web-Based Tasks

We consider the WebVoyager dataset (He et al., 2024), where $X$ is a task for an AI agent with access to the web. These tasks simulate common user objectives, such as locating a research paper on arXiv, reserving a hotel on booking.com, or searching for a specific place on Google. Each generation $\mathbf{Y}_t$ is itself a trajectory of actions taken by the

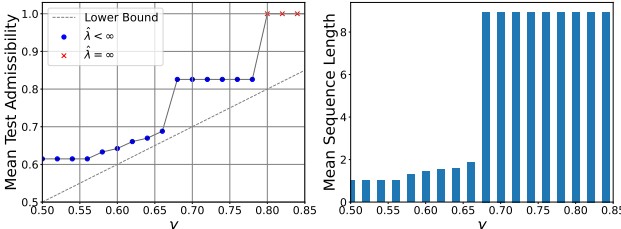

*Figure 5.* Results on test data for the agentic AI tasks.

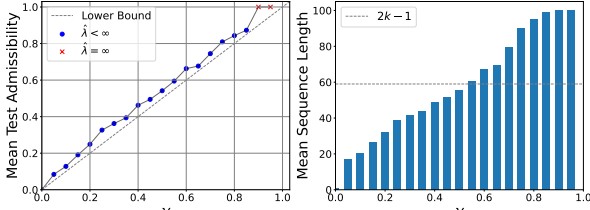

*Figure 6.* Results on test data for the random forest.

agent, with each $\mathbf{Y}_t$ being generated independently. We use the selection function from the first row of Table 1 (with $\texttt{accum} = \max$), where the score function $S^{\uparrow}(x, \mathbf{y}_t)$ is obtained by instructing an LLM to assess the percentage at which $\mathbf{y}_t$ has completed task $x$. The admissibility function is a binary indicator of whether the task is successfully completed by the end of the trajectory (determined by an LLM judge). Thus, the conformal guarantee ensures that the selected sequence contains at least one successful trajectory with probability at least $\gamma$ (as assessed by the LLM judge, which He et al. (2024) tested thoroughly).

We follow He et al. (2024) to instantiate the AI agent, and report results in Figure 5. While the agent achieves a single-attempt success rate near 60%, Conf-Gen formally guarantees above 65% success averaging under two attempts, rising to nearly 80% when more are permitted. More details are provided in Appendix E.4.

We believe that this task could help enhance the reliability of AI agents. For example, the agent could execute all its trajectories in a simulated environment, so that each trajectory corresponds to a plan. The user could then be shown the corresponding subset of selected trajectories and choose which plan to execute, if any. Such a procedure could help ensure that, for example, an agent tasked with booking a hotel online does not incur a financial loss when it makes a mistake.

### 6.5. Conformal Random Forests

To further demonstrate the generality of Conf-Gen, we consider a task in supervised learning beyond the scope of generative models. We assume access to a pre-trained random forest classifier with $T$ trees, where $X$ is a feature vector, $Y_t$ contains the prediction from the $t$-th tree, and $Y_{\text{GT}}$ is the ground truth label. Our selection function outputs the smallest subset of trees whose accumulated score (with $\texttt{accum} = \texttt{sum}$) exceeds $\lambda$, where the score function $S^{\uparrow}$ is given by the weighted tree sample count. The selected trees are then used to make a prediction, ignoring the other trees. Admissibility is defined via the binary variable $A'_t$, which indicates whether the $t$-th tree correctly predicts $Y_{\text{GT}}$ for $X$, and the aggregation function $\texttt{agg}(\mathbf{a}') = \mathbb{1}(\sum_{a' \in \mathbf{a}'} a' \geq k)$ for a user-specified $k$. Consequently, Conf-Gen guarantees that the probability of the selected subset containing at least

$k$ trees which make the correct prediction is at least $\gamma$. This guarantee is particularly meaningful when the selected set contains at most $2k - 1$ trees, as it then ensures the correctness of the majority-vote prediction.

While defining admissibility directly as the correctness of the final prediction might seem more intuitive—as the resulting conformal guarantee would then ensure that the prediction is correct with probability at least $\gamma$—such a formulation fails to ensure the monotonicity of $\lambda \mapsto A(\lambda)$, even in conditional expectation. Our chosen definition of $A$ avoids this issue, and satisfies the necessary theoretical properties while still providing a meaningful conformal guarantee.

We pre-train a random forest with $T = 100$ trees on the Click_prediction_small dataset (Coutinho, 2022) and set $k = 30$. The results in Figure 6 demonstrate that the conformal guarantee is empirically satisfied and that the average number of selected trees remains below $2k - 1 = 59$ across a wide range of $\gamma$ values. Additional details, experiments on more datasets, and ablations over $k$ are provided in Appendix E.5.

## 7. Conclusion and Future Work

In this work, we presented Conf-Gen, a framework that extends CRC to generative tasks while relaxing its underlying theoretical assumptions. We also provide a Python package to facilitate the deployment of Conf-Gen across diverse settings. Our experiments demonstrate that Conf-Gen not only outperforms current state-of-the-art conformal methods in open-domain question answering but also establishes a foundation for a wide array of novel conformal applications.

We hope that our work will catalyze the adoption of conformal guarantees for UQ in generative modelling. Future work may focus on refining the data-generating processes, score functions, and selection functions we proposed for our considered tasks. Beyond these potential technical improvements, exploring new use cases for Conf-Gen remains a compelling research direction.

## Impact Statement

Our work presents a step towards improved uncertainty quantification in generative models. These type of improve-

ments could have various societal consequences. On the positive side, they could increase the overall reliability of AI systems, help deploy them in safety-critical applications, and increase trust in them; indeed, these are some of the key motivations behind our work. On the negative side, uncertainty quantification can also improve the reliability of malicious AI applications. Overall, we believe that uncertainty quantification for generative models is a meaningful research topic precisely because it is likely to have more positive than negative impact. Although we do not believe that any of the experiments in this paper are likely to have direct societal consequences, future applications might.

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

## A. Review of Existing Upper Bounds

As mentioned in the main text, there are accompanying results to those in Theorem 1 and Theorem 2 which provide upper bounds. We begin by reviewing the upper bound for CP.

**Theorem 4** (Angelopoulos et al. (2024a)). *Under the conditions of Theorem 1, we have that*

$$\mathbb{P}\left(Y_{\text{GT}}^{(n+1)} \in \mathcal{C}_{\hat{\lambda}}\left(X^{(n+1)}\right)\right) \leq \gamma + \frac{1}{n+1} + \epsilon, \tag{13}$$

*where $\epsilon$ is the probability of the $(n+1)^{th}$ score being equal to another score, i.e.,*

$$\epsilon := \mathbb{P}\left(S^{\downarrow}\left(X^{(j)}, Y_{\text{GT}}^{(j)}\right) = S^{\downarrow}\left(X^{(n+1)}, Y_{\text{GT}}^{(n+1)}\right) \text{ for some } j \in \{1, \ldots, n\}\right). \tag{14}$$

The first such upper bound was proved by Lei et al. (2018) under the assumption that the scores follow a continuous joint distribution, meaning that $\epsilon = 0$.

CRC also admits an upper bound, which requires the notion of a càdlàg function and its jumps.

**Definition 2.** *Consider $\Lambda = [\lambda_{\min}, \lambda_{\max}]$. The function $f : \Lambda \to \mathbb{R}$ is called càdlàg if it is right-continuous and $\lim_{\epsilon \to 0^+} f(\lambda - \epsilon)$ exists for every $\lambda \in (\lambda_{\min}, \lambda_{\max}]$. In this case, the jump of $f$ at $\lambda \in \Lambda$ is defined as*

$$h(f, \lambda) := \begin{cases} f(\lambda) - \lim_{\epsilon \to 0^+} f(\lambda - \epsilon), & \text{if } \lambda \in (\lambda_{\min}, \lambda_{\max}]. \\ f(\lambda), & \text{if } \lambda = \lambda_{\min}. \end{cases} \tag{15}$$

Note that any right-continuous, non-decreasing function is càdlàg, so that under the assumptions of Theorem 2, $U^{(i)}$ is càdlàg almost surely for $i = 1, \ldots, n+1$. We now state the upper bound for CRC.

**Theorem 5** (Angelopoulos et al. (2024b)). *Assume that the requirements of Theorem 2 hold. Additionally, assume that $U : 2^{\mathcal{Y}_{\text{GT}}} \times \mathcal{Y}_{\text{GT}} \to [0, u_{\max}]$, that for $i = 1, \ldots, n+1$, the random functions $U^{(i)}$ are i.i.d., and that $\mathbb{P}(h(U^{(i)}, \lambda) > 0) = 0$ for every $\lambda \in \Lambda$. Then,*

$$\mathbb{E}\left[U^{(n+1)}(\hat{\lambda})\right] \leq \gamma + \frac{2u_{\max}}{n+1}. \tag{16}$$

Note that originally, Angelopoulos et al. (2024b) provided a lower bound for a loss function; the restatement we wrote above corresponds to the case where the loss is given by $u_{\max}$ minus the utility function. Note also that, rather than using $\gamma$ directly, Angelopoulos et al. (2024b) write the bound in terms of $\alpha := u_{\max} - \gamma$. Once again, writing the bound for CRC in terms of a utility function rather than a loss function facilitates comparing between CRC and CG. Note also that the assumption that $\mathbb{P}(h(U^{(i)}, \lambda) > 0) = 0$ for every $\lambda \in \Lambda$ is analogous to the scores having a continuous distribution in CP. Lastly, we also note that when $U(\mathcal{C}, y) = \mathbb{1}(y \in \mathcal{C})$, the bound obtained in CRC is weaker than the corresponding bound in CP by a factor of 2; this is discussed in more detail by Angelopoulos et al. (2024b).

## B. Proof of Theorem 3

We begin by restating Definition 1 and Theorem 3 for convenience.

**Definition 1.** *We say that an instance of Conf-Gen is $\gamma$-sensible if the following properties hold:*

*(a) The functions $A^{(1)}, \ldots, A^{(n+1)}$ are exchangeable.*

*(b) For every $\gamma', \gamma'' \geq 0$, if $\lambda' = \inf \mathcal{V}(\mathbf{D}_{:n}, \gamma')$ and $\lambda'' = \inf \mathcal{V}(\mathbf{D}_{:n+1}, \gamma'')$, then:*

    *(b1) The map given by $\lambda \mapsto \mathbb{E}[A^{(n+1)}(\lambda) \mid \lambda', \lambda'']$ is non-decreasing on $\Lambda$, almost surely.*

    *(b2) $\lambda' \in \mathcal{V}(\mathbf{D}_{:n}, \gamma')$ and $\lambda'' \in \mathcal{V}(\mathbf{D}_{:n+1}, \gamma'')$, almost surely.*

*(c) $A(X, \mathbf{C}_{\infty}(X, \mathbf{Y}), Y_{\text{GT}}) \geq \gamma$, almost surely.*

**Theorem 3.** *Consider a $\gamma$-sensible instance of Conf-Gen. Then, for $\hat{\lambda}$ as defined in Equation 6,*

$$\mathbb{E}\left[A^{(n+1)}(\hat{\lambda})\right] \geq \gamma.$$

*Additionally, if $A : \mathcal{X} \times \mathcal{S} \times \mathcal{Y}_{\text{GT}} \to [0, a_{\max}]$, then*

$$\mathbb{E}\left[A^{(n+1)}(\hat{\lambda})\right] \leq \gamma + \frac{a_{\max}}{n+1} + \mathbb{E}[H],$$

*where $\lambda^{**} := \inf \mathcal{V}(\mathbf{D}_{:n+1}, \gamma + \frac{a_{\max}}{n+1})$ and*

$$H := \bar{A}_{n+1}(\lambda^{**}) - \sup\left\{\bar{A}_{n+1}(\lambda) : \lambda \in \Lambda, \lambda < \lambda^{**}\right\} \vee 0.$$

*Proof.* We first prove the lower bound. Let

$$\lambda^* := \inf \mathcal{V}\left(\mathbf{D}_{:n+1}, \gamma\right). \tag{17}$$

Note that $\lambda^*$ cannot be computed in practice since we do not have access to the $(n+1)$-th datapoint; we merely use this quantity as a theoretical construct in our proof. Since $\lambda^* \in \mathcal{V}(\mathbf{D}_{:n+1}, \gamma) \subseteq \Lambda$, almost surely, then $\mathbb{P}(\lambda^* \in \Lambda) = 1$, so that $A^{(i)}(\lambda^*)$ is almost surely well-defined for $i = 1, \ldots, n+1$. Additionally, due to the considered instance of Conf-Gen being $\gamma$-sensible, $\bar{A}_{n+1}(\lambda^*) \geq \gamma$, almost surely. Note that $\lambda^*$ is a symmetric and deterministic function of $A^{(1)}, \ldots, A^{(n+1)}$, which are exchangeable by assumption. By standard properties of exchangeability (Angelopoulos et al., 2024a), it follows that

$$\mathbb{E}\left[A^{(n+1)}(\lambda^*)\right] = \mathbb{E}\left[A^{(i)}(\lambda^*)\right] \tag{18}$$

for every $i = 1, \ldots, n+1$. We then have that

$$\mathbb{E}\left[A^{(n+1)}(\lambda^*)\right] = \frac{1}{n+1}\sum_{i=1}^{n+1}\mathbb{E}\left[A^{(i)}(\lambda^*)\right] = \mathbb{E}\left[\frac{1}{n+1}\sum_{i=1}^{n+1}A^{(i)}(\lambda^*)\right] = \mathbb{E}\left[\bar{A}_{n+1}(\lambda^*)\right] \geq \gamma. \tag{19}$$

We now want to show that $\lambda^* \leq \hat{\lambda}$, almost surely. There are two cases: either $\hat{\lambda} = \infty$, or $\hat{\lambda} < \infty$. In the former case, the inequality clearly holds. Otherwise, if $\hat{\lambda} < \infty$, we have that $\hat{\lambda} \in \mathcal{V}(\mathbf{D}_{:n}, \frac{n+1}{n}\gamma) \setminus \{\infty\}$, and thus

$$\frac{1}{n}\sum_{i=1}^{n}A^{(i)}(\hat{\lambda}) \geq \frac{n+1}{n}\gamma \implies \sum_{i=1}^{n}A^{(i)}(\hat{\lambda}) \geq (n+1)\gamma \tag{20}$$

$$\implies \sum_{i=1}^{n+1}A^{(i)}(\hat{\lambda}) \geq (n+1)\gamma \tag{21}$$

$$\implies \frac{1}{n+1}\sum_{i=1}^{n+1}A^{(i)}(\hat{\lambda}) \geq \gamma \tag{22}$$

$$\implies \hat{\lambda} \in \mathcal{V}(\mathbf{D}_{:n+1}, \gamma), \tag{23}$$

almost surely, where the second implication follows from the non-negativity of $A$. Since $\lambda^* = \inf \mathcal{V}(\mathbf{D}_{:n+1}, \gamma)$, it follows that $\lambda^* \leq \hat{\lambda}$ also holds in this case. Now, we denote $B(\lambda) := \mathbb{E}[A^{(n+1)}(\lambda) \mid \hat{\lambda}, \lambda^*]$. Lastly, since by assumption $B$ is almost surely non-decreasing, we have that

$$\mathbb{E}\left[A^{(n+1)}(\hat{\lambda})\right] = \mathbb{E}\left[\mathbb{E}\left[A^{(n+1)}(\hat{\lambda}) \mid \hat{\lambda}, \lambda^*\right]\right] = \mathbb{E}[B(\hat{\lambda})] \geq \mathbb{E}[B(\lambda^*)] \tag{24}$$

$$= \mathbb{E}\left[\mathbb{E}\left[A^{(n+1)}(\lambda^*) \mid \hat{\lambda}, \lambda^*\right]\right] = \mathbb{E}\left[A^{(n+1)}(\lambda^*)\right] \geq \gamma, \tag{25}$$

where the second and third equalities follow from $B(\lambda)$ being $\sigma(\hat{\lambda}, \lambda^*)$-measurable for every $\lambda \in \Lambda$, where $\sigma(\hat{\lambda}, \lambda^*)$ denotes the $\sigma$-algebra generated by $\hat{\lambda}$ and $\lambda^*$. This concludes the proof of the lower bound.

We now prove the upper bound and assume that $A$ is upper-bounded by $a_{\max}$. Note that, by definition of $\lambda^{**}$,

$$\forall \lambda < \lambda^{**}, \quad \frac{1}{n+1}\sum_{i=1}^{n+1}A^{(i)}(\lambda) < \gamma + \frac{a_{\max}}{n+1}. \tag{26}$$

Therefore, for every $\lambda < \lambda^{**}$,

$$\frac{1}{n+1}\sum_{i=1}^{n+1} A^{(i)}(\lambda^{**}) - \frac{1}{n+1}\sum_{i=1}^{n+1} A^{(i)}(\lambda) > \frac{1}{n+1}\sum_{i=1}^{n+1} A^{(i)}(\lambda^{**}) - \gamma - \frac{a_{\max}}{n+1}, \tag{27}$$

which then implies that

$$H \geq \frac{1}{n+1}\sum_{i=1}^{n+1} A^{(i)}(\lambda^{**}) - \gamma - \frac{a_{\max}}{n+1} \tag{28}$$

holds almost surely. It follows that

$$\mathbb{E}\left[A^{(n+1)}(\lambda^{**})\right] = \mathbb{E}\left[\frac{1}{n+1}\sum_{i=1}^{n+1} A^{(i)}(\lambda^{**})\right] \leq \gamma + \frac{a_{\max}}{n+1} + \mathbb{E}[H], \tag{29}$$

where the first equality holds due to exchangeability (this is analogous to the reasoning for the lower bound). Now, note that the following implications hold:

$$\frac{1}{n+1}\sum_{i=1}^{n+1} A^{(i)}(\lambda^{**}) \geq \gamma + \frac{a_{\max}}{n+1} \implies \frac{1}{n}\sum_{i=1}^{n+1} A^{(i)}(\lambda^{**}) \geq \frac{(n+1)\gamma}{n} + \frac{a_{\max}}{n} \tag{30}$$

$$\implies \frac{1}{n}\sum_{i=1}^{n} A^{(i)}(\lambda^{**}) \geq \frac{(n+1)\gamma}{n} + \frac{1}{n}\left(a_{\max} - A^{(n+1)}(\lambda^{**})\right) \tag{31}$$

$$\implies \frac{1}{n}\sum_{i=1}^{n} A^{(i)}(\lambda^{**}) \geq \frac{(n+1)\gamma}{n} \implies \lambda^{**} \in \mathcal{V}\left(\mathbf{D}_{:n}, \frac{n+1}{n}\gamma\right). \tag{32}$$

Following once again an analogous reasoning to the one for the lower bound, we have that $\hat{\lambda} \leq \lambda^{**}$ holds almost surely and that

$$\mathbb{E}\left[A^{(n+1)}(\hat{\lambda})\right] = \mathbb{E}\left[\mathbb{E}\left[A^{(n+1)}(\hat{\lambda}) \mid \hat{\lambda}, \lambda^{**}\right]\right] \leq \mathbb{E}\left[\mathbb{E}\left[A^{(n+1)}(\lambda^{**}) \mid \hat{\lambda}, \lambda^{**}\right]\right] \tag{33}$$

$$= \mathbb{E}\left[A^{(n+1)}(\lambda^{**})\right] \leq \gamma + \frac{a_{\max}}{n+1} + \mathbb{E}[H]. \tag{34}$$

$\square$

## C. Additional Discussion of Conf-Gen: Assumptions and Comparison to CRC

We now provide an extended discussion of the assumptions of Conf-Gen, as well as its similarities and differences with CRC.

First, we discuss the monotonicity assumption in Definition 1, which we note we only use with $\gamma' = \frac{n+1}{n}\gamma$ and $\gamma'' = \gamma$ when proving the lower bound, i.e., that $\lambda \mapsto \mathbb{E}[A^{(n+1)}(\lambda) \mid \hat{\lambda}, \lambda^*]$ is non-decreasing, almost surely. This assumption is used in the proof of Theorem 3 in Equation 24 to establish that $\mathbb{E}[A^{(n+1)}(\hat{\lambda})] \geq \mathbb{E}[A^{(n+1)}(\lambda^*)]$ follows from $\lambda^* \leq \hat{\lambda}$. Note that the assumption of monotonicity only in (total) expectation—i.e., that $\lambda \mapsto \mathbb{E}[A^{(n+1)}(\lambda)]$ is non-decreasing—is not enough to conclude the desired inequality holds, due to $\hat{\lambda}$ not being independent from $A$ and $\mathbf{C}_\lambda$, and $\lambda^*$ also not being independent from $(X^{(n+1)}, \mathbf{Y}^{(n+1)}, Y_{\mathrm{GT}}^{(n+1)})$. Note also that the assumption of monotonicity in total expectation follows from monotonicity in conditional expectation. Thus, our monotonicity condition is stronger than monotonicty in total expectation, but weaker than $\lambda \mapsto A^{(n+1)}(\lambda)$ being non-decreasing almost surely, as required by CRC. Our condition can be understood as strengthening the assumption of monotonicity in total expectation by requiring that knowledge of $\hat{\lambda}$ and $\lambda^*$ must not alter the distribution of $A$, $\mathbf{C}_\lambda$, and $(X^{(n+1)}, \mathbf{Y}^{(n+1)}, Y_{\mathrm{GT}}^{(n+1)})$ so drastically so as to violate monotonicity in expectation. Intuitively, we can expect monotonicity in conditional expectation to hold whenever monotonicity in total expectation holds and the dependence between $A^{(n+1)}(\lambda)$ and $(\hat{\lambda}, \lambda^*)$ is sufficiently weak.

Second, the assumption in Conf-Gen that $A$ be non-negative can be understood as simply requiring $A$ to be lower-bounded (and the same is true for CRC). More precisely, assume we have an admissibility function $\tilde{A}$ which is lower-bounded by

$a_{\min} \in \mathbb{R}$ rather than 0, and say we are given a scalar admissibility target $\tilde{\gamma} \geq a_{\min}$. We can then define $A := \tilde{A} - a_{\min}$ and $\gamma := \tilde{\gamma} - a_{\min}$, and by non-negativity of $A$, Theorem 3 holds. Adding $a_{\min}$ to both sides of Equation 8 then yields the desired conformal guarantee:

$$\mathbb{E}\left[\tilde{A}\left(X^{(n+1)}, \mathbf{C}_{\hat{\lambda}}\left(X^{(n+1)}, \mathbf{Y}^{(n+1)}\right), Y_{\text{GT}}^{(n+1)}\right)\right] \geq \tilde{\gamma}. \tag{35}$$

Note that in this case, $\hat{\lambda}$ as defined in Equation 6 can be re-written in terms of $\tilde{A}$ and $\tilde{\gamma}$ as

$$\hat{\lambda} = \inf\left\{\lambda \in \Lambda \;\middle|\; \frac{1}{n}\sum_{i=1}^{n}\tilde{A}\left(X^{(i)}, \mathbf{C}_{\lambda}\left(X^{(i)}, \mathbf{Y}^{(i)}\right), Y_{\text{GT}}^{(n+1)}\right) \geq \frac{n+1}{n}\tilde{\gamma} - \frac{a_{\min}}{n}\right\}. \tag{36}$$

Third, the proof of Theorem 3 requires $\hat{\lambda} \in \mathcal{V}(\mathbf{D}_{:n}, \frac{n+1}{n}\gamma)$ and $\lambda^{*} \in \mathcal{V}(\mathbf{D}_{:n}, \gamma)$ to hold almost surely: this happens whenever $\lambda \mapsto A(X, \mathbf{C}_{\lambda}(X, \mathbf{Y}), Y_{\text{GT}})$ is right-continuous and $\Lambda$ is a closed interval, as required in CRC—yet it can also hold in other settings, e.g., if $\Lambda$ is finite.

Fourth, Conf-Gen deals with $\Lambda$ more cleanly than CRC by making it clear that this set can be unbounded and that it need not be an interval.

Lastly, the upper bound in Equation 9 improves upon the corresponding bound for CRC from Equation 16 in various ways: it has the same weaker assumptions than CRC for the lower bound, it does not require independence nor the càdlàg property, and its use of $\mathbb{E}[H]$ is somewhat analogous to $\epsilon$ in the CP upper bound (Theorem 4), whereas the CRC bound (Theorem 5) has no such analogue. To see that our bound indeed generalizes the CRC one, note that if the functions $A^{(i)}$ are càdlàg, then $H \leq h(\bar{A}_{n+1}, \lambda^{**})$, and if additionally these functions are independent and $\mathbb{P}(h(A^{(i)}, \lambda) > 0) = 0$ for every $\lambda \in \Lambda$ (as required by CRC), then no more than a single jump can happen at the same time, which yields that $h(\bar{A}_{n+1}, \lambda^{**}) \leq \frac{a_{\max}}{n+1}$ almost surely (see Angelopoulos et al. (2024b) for a formal derivation of this fact). In other words, our bound implies the CRC bound when we add the càdlàg and independence assumptions used by CRC, yet it remains valid without them. For example, if the functions $A^{(i)}$ are continuous and $\mathbb{P}(\lambda^{**} = \inf \Lambda) = 0$, then $H = 0$ almost surely, in which case our upper bound improves to $\gamma + \frac{a_{\max}}{n+1}$.

# D. Additional Examples

Table 2 is an extended version of Table 1, including the discussion below and the names of each sequence selector as it appears in our Python package. We invite the reader to consult the package documentation for a complete list of features and configurations beyond those highlighted in Table 2.

Here we provide additional examples of Conf-Gen configurations beyond those presented in Section 4.3. First, note that we can always apply a post-processing step to the selection function outputs from Table 1; formally this means applying a function to this output. For example, in the experiment from Section 6.1, the selection function is $f(\mathbf{y}_{:\tau(x,\mathbf{y},\lambda)})$, where $f : \mathcal{Y}^{*} \to \mathcal{Y}^{*}$ is a deterministic function which removes all exact duplicates from its input sequence. More complex choices of post-processing functions are also possible. For instance, we might want to remove all "semantic duplicates", which could be achieved by outputting $F(\mathbf{y}_{:\tau(x,\mathbf{y},\lambda)})$, where $F$ is an LLM call instructing the LLM to remove responses with the same meaning, even if they are not identical. Note that in general, the output of the selection function $\mathbf{C}_{\lambda}(x, \mathbf{y})$ in this example need not be a subsequence of $\mathbf{y}$, although the finite image property remains nonetheless valid.

We also mentioned in Section 5 that the conformal summarization method of Kuwahara et al. (2025) is an instance of Conf-Gen. Like conformal factuality, this corresponds to the third row of Table 2, except this time the aggregation function is different. Here, there is no ground truth variable, and we define $A''_t(x, y_t)$ as an indicator of whether sentence $y_t$ is relevant within document $x$. We also define $A'_t(x, \mathbf{y})$ as $\frac{1}{N(x)}\sum_t A''_t(x, y_t)$, where $N(x)$ is the number of relevant sentences in document $x$. The aggregation function is then given by

$$\text{agg}(\mathbf{a}') = \mathbb{1}\left(\sum_{a' \in \mathbf{a}'} a' \geq \beta\right). \tag{37}$$

Note that Conf-Gen also allows us to easily modify this setup. For example, if we wanted the output to be a proper summary rather than a subset of important sentences, we could change the output from $\{y_t | S_t^{\uparrow} > \lambda\}$ to $F(\{y_t | S_t^{\uparrow} > \lambda\})$, where $F$ is an LLM call where the LLM is instructed to summarize all the sentences given to it as input.

*Table 2.* Representative configurations of Conf-Gen.

| Name | Selection function | | Compatible with | | |
| --- | --- | --- | --- | --- | --- |
| | $\mathbf{C}_\lambda(x, \mathbf{y})$ | accum | agg | Partial generation | Post-processing |
| RunningMax / RunningSum | $\mathbf{y}_{:\tau(x,\mathbf{y},\lambda)}$ | max / sum | max / Equation 37 | Yes | Yes |
| BelowLambda | $\{y_t \mid S_t^\downarrow \le \lambda\}$ | N/A | max / Equation 37 | No | Yes |
| AboveLambda | $\{y_t \mid S_t^\uparrow > \lambda\}$ | N/A | min | No | Yes |
| RunningMaxSingle | $y_{\tau(x,\mathbf{y},\lambda)}$ | max | max / min | Yes | Yes |
| SmallestSubsetSum | Equation 38 | sum | max / Equation 37 | No | Yes |

Lastly, in Section 6.5 we used a selection function which is not displayed in Table 1: the smallest subset such that the accumulated score exceeds $\lambda$. Formally, for a sequence $\mathbf{y}$ with corresponding scores $\mathbf{S}^\uparrow$, this selection function is given by

$$\mathbf{C}_\lambda(x, \mathbf{y}) \in \underset{\mathcal{U} \subseteq \mathbf{y} \mid \sum_{s^\uparrow \in \mathbf{s}_{\mathcal{U}}^\uparrow} S^\uparrow > \lambda}{\arg \min} |\mathcal{U}| \tag{38}$$

whenever the $\arg \min$ is not empty, and $\mathbf{C}_\lambda(x, \mathbf{y}) = \mathbf{y}$ otherwise, where $\mathbf{S}_{\mathcal{U}}^\uparrow$ contains the entries in $\mathbf{S}^\uparrow$ corresponding to $\mathcal{U}$. Operationally, this selection function is equivalent to the first row of Table 2, except the sequence $\mathbf{y}$ is sorted beforehand on decreasing order of its corresponding scores (and the output is a set rather than a sequence). Note that the sorting operation requires access to the entire sequence $\mathbf{y}$, and thus this selection function is no longer compatible with partial generation. Note also that this selection function behaves similarly to the second row from Table 2, except it does not include all entries with matching scores at once. As an example, consider a sequence $\mathbf{y} = (y_1, y_2, y_3)$ with corresponding scores $\mathbf{s}^\uparrow = (1, 1, 2)$, and assume that $\mathbf{s}^\downarrow = -\mathbf{s}^\uparrow$. This selection function would produce $\{y_3\}$, $\{y_3, y_1\}$, and $\{y_3, y_2, y_1\}$ (assuming the sorting algorithm produces $(y_3, y_1, y_2)$ and not $(y_3, y_2, y_1)$), with the jumps happening at $\lambda = 2, 3$. The selection function from the second row of Table 2 would produce $\emptyset$, $\{y_3\}$, and $\{y_3, y_2, y_1\}$, with the jumps happening at $\lambda = -2, -1$: this latter selection function always outputs $y_1$ and $y_2$ together as they have the same score.

# E. Experiments

## E.1. Conformal Open-Domain Question Answering

**Dataset Splits.** The TriviaQA dataset (Joshi et al., 2017) contains trivia questions such as *"What is Harrison Ford's real name?"* or *"What is Mel Gibson's middle name?"* for which there exist a small number of correct answers (often only one). We use the same TriviaQA dataset used by Quach et al. (2024) composed of $17,944$ trivia questions and their answers. Quach et al. (2024) decompose their dataset as follows: $2,000$ questions are used for training, $2,000$ questions are used for validation, and the rest is used for test. We use their training set as our calibration set, and our test set is the same as theirs.

**Protocol.** We follow the experimental protocol from Quach et al. (2024) by using LLaMa-13B embeddings (Touvron et al., 2023). For each question $x$, we sample $T = 20$ answers. We define $S^\uparrow(x, y)$ as the likelihood function of the LLaMA model with length-normalization (Wu et al., 2016) for the question $x$ and the answer $y$. The admissibility function $A'(x, y, y_{\text{GT}})$ is 1 if $y$ matches $y_{\text{GT}}$, and 0 otherwise.

**Sampling Strategies.** In the main paper, we consider that our sampling strategy is *i.i.d.*, which means that the LLM can generate the same answer multiple times. As an ablation study, we propose here an alternative sampling strategy (that we call *diverse*) that forces the LLM to generate an answer that is different from all the answers that were previously generated. To this end, and due to the nature of the dataset and of the ground truth answers that are often short, we found that an easily reproducible way of generating different answers was to add previously generated answers to the list of *bad words*.[5] When explicitly asked in the prompt to generate different answers, LLaMa tends to generate the same answer multiple times because of the high certainty of the model for its first generated answer; this is why we used the list of bad words instead.

**Results.** Figure 2 illustrates the average test admissibility score and the average number of answers generated for each question by Quach et al. (2024) with the *max scoring functions* that they propose, and by our method with the selection function from the first row of Table 1 with accum = max but with duplicates removed, as a function of $\gamma \in [0.62, 0.86]$ (as done by Quach et al. (2024)). As discussed in the main paper, our approach is cheaper at inference time since it requires

---

[5]See https://huggingface.co/docs/transformers/v4.46.0/en/internal/generation_utils for details. In practice, this corresponds to a list of token ids that are not allowed to be generated.

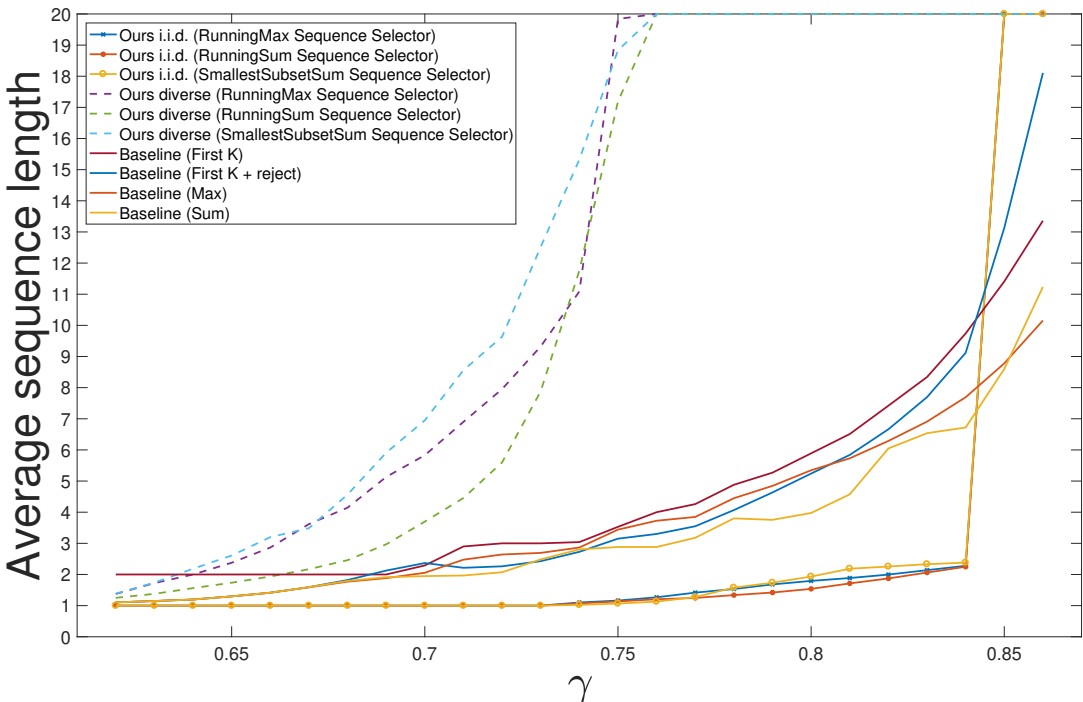

*Figure 7.* Average sequence length as a function of $\gamma$ with different sampling strategies. Our proposed strategy results in shorter average sequence lengths than the baseline, resulting in a smaller number of LLM calls.

fewer LLM calls for a given $\gamma \leq 0.84$. When $\gamma \geq 0.85$, the value of our $\hat{\lambda}$ is $+\infty$, and the size of our sequence becomes as large as possible. Quach et al. (2024) avoid this degenerate case by using different filters with the Pareto testing method (Laufer-Goldshtein et al., 2023) to cross-validate hyperparameters that our approach does not use, but this cross-validation process also incurs additional cost during calibration, and it results in larger sets when $\gamma \leq 0.84$.

Figure 7 illustrates the average sequence length of the baseline and of our method with *i.i.d.* sampling and with the *diverse* strategy described above with different sequence selectors described in Table 2. Since the first answer generated by the model is often correct and there are few possible correct answers in this task, the *diverse* sampling strategy underperforms both the strategy with *i.i.d.* sampling and the baseline. This demonstrates that the choice of the *sequence generation* strategy is a crucial step of our approach and may be impacted by the nature of the task and of the dataset. On the other hand, the choice of the sequence selector function has less impact on the performance in this task. We also report the results obtained by the baseline with different scoring functions called *First-K*, *First-K+reject*, *Max*, and *Sum* (see Quach et al., 2024, Section 5.2).

### E.2. Conformal Generation of Non-Memorized Images

**Score Function.** We now specify the score function we used for the experiment in Section 6.2 in more detail. Recall that diffusion models (Sohl-Dickstein et al., 2015; Ho et al., 2020; Song et al., 2021) such as Stable Diffusion generate an image by first sampling Gaussian noise and iteratively refining the noise into an image. Since Stable Diffusion is a latent diffusion model, this refinement happens in latent space, and a decoder $g$ must be applied to obtain an image from a latent. Formally, the $t$-th generation does not only contain the final generated latent, but the entire trajectory of noisy latents used by the diffusion model during generation: $\mathbf{Y}_t = (Y_t(1), \ldots, Y_t(m))$, where $Y_t(1)$ corresponds to the final generated latent (i.e., $g(Y_t(1))$ is the final generated image), with each $Y_t(j)$ becoming progressively nosier as $j$ increases, and with $Y_t(m)$ corresponding to pure Gaussian noise. Recall that diffusion models provide a score function $s$,[6] and the classifier guidance term corresponding to $Y_t(j)$ is

$$s\left(Y_t(j), j, X_t\right) - s\left(Y_t(j), j, \emptyset\right), \tag{39}$$

---

[6]This function corresponds to the Stein score, and has nothing to do with our use of the term "score function".

i.e., it is the difference between the score function conditioned on the input prompt $X_t$, and the score function without conditioning on any prompt. Our score function is given by $S_t^\uparrow = -S_t^\downarrow$, where

$$S_t^\downarrow = S^\downarrow(X_t, \mathbf{Y}_t) = \frac{1}{m} \sum_{j=1}^{m} \sigma^2(j) \| s\left(Y_t(j), j, X_t\right) - s\left(Y_t(j), j, \emptyset\right) \|_2^2, \tag{40}$$

where $\sigma(j)$ is a known coefficient from the diffusion process. As mentioned in the main text, Wen et al. (2023) found this score indicative of memorization, and we used the procedure proposed by Ross et al. (2025), which we now describe. $\mathbf{Y}_1$ is produced directly from $X_1$, and then the gradient of $S^\downarrow(X_1, \mathbf{Y}_1)$ is computed for every token in $X_1$. The token achieving the largest gradient norm is selected, and an LLM is instructed to produce $X_2$ by changing the selected token from $X_1$, while keeping the entire prompt semantically meaningful. Then, $\mathbf{Y}_2$ is generated using $X_2$ as the prompt, and this process is repeated sequentially to produce $X_3$ based on $S^\downarrow(X_2, \mathbf{Y}_2)$, and so on. Ross et al. (2025) proposed this procedure as a way to generate a sequence of progressively less memorized images; our work can be understood as conformalizing the choice of which of these images to select. Ross et al. (2025) found that a different score based on local intrinsic dimension estimates (Kamkari et al., 2024; Leung et al., 2025) performed slightly better than the one in Equation 40, but this score is more computationally taxing to compute and resulted in out-of-memory errors on our hardware.

**Dataset.** We used the code of Ross et al. (2025), which uses the samples that Webster (2023) identified as memorized by Stable Diffusion v1.5. The codebase does not directly provide images, but rather URLs containing the images. Some of the URLs were no longer valid, and some of the valid URLs contained duplicate images. After de-duplication, we were left with 52 images, which we split into a calibration and a test set each containing 26 images. For each one of these ground truth memorized images $Y_{\text{GT}}^{(i)}$ and each corresponding generated image $g(Y_t^{(i)}(1))$, we asked 10 human evaluators to choose one of three options:[7] $(a)$ "bad", meaning that the generated image is a copy or near-copy of the real image; $(b)$ "medium", meaning that the generated image is "too different" or completely unrelated to the real one; and $(c)$ "good", meaning that the generated image is similar in content and style to the real one, but not close enough to be a copy. The admissibility $A_t'$ is then given by the percentage of evaluators who assessed the corresponding generated image as "good" or "medium". In Figure 8, Figure 9, and Figure 10, we display examples shown to the human evaluators, corresponding to a "bad", a "medium", and a "good" generation, respectively.

**Discussion.** Note that this experiment is an example where the function $\lambda \mapsto A(\lambda)$ is not non-decreasing, almost surely: it is possible, however unlikely, that by changing more tokens in the prompt, the corresponding generated image becomes more memorized. Nonetheless, since this event should be expected to have very low probability, it is reasonable to believe that monotonicity still holds in conditional expectation, as required by Definition 1. This example underscores the practical relevance of our relaxed theoretical framework over the standard requirements of CRC.

**Human Evaluators.** We used the Prolific platform to recruit our anonymous human evaluators. We recruited a total of 40 participants; the only restrictions we enforced were self-reported fluency in English (in order to understand instructions), and a minimum of a technical college degree, so as to maximize the probability of true English fluency. We manually reviewed every image before running the experiment to ensure no disturbing content would be shown to the participants. We paid an average hourly rate above £12, which is significantly more than the Prolific-recommended amount, £9. Some institutions have an institutional review board (IRB) which sets the ethical standards that its members must follow when conducting experiments with humans, and whose approval is required prior to running such experiments. Our institution has no IRB, and we thus followed the aforementioned steps to ensure our experiment was run ethically.

---

[7]Note that this is an example where the elements of $\mathbf{Y}$ do not live in the same space as $Y_{\text{GT}}$: the former are latents, and the later is an image. In other words, this example highlights that Conf-Gen remains valid when $\mathcal{Y} \neq \mathcal{Y}_{\text{GT}}$.

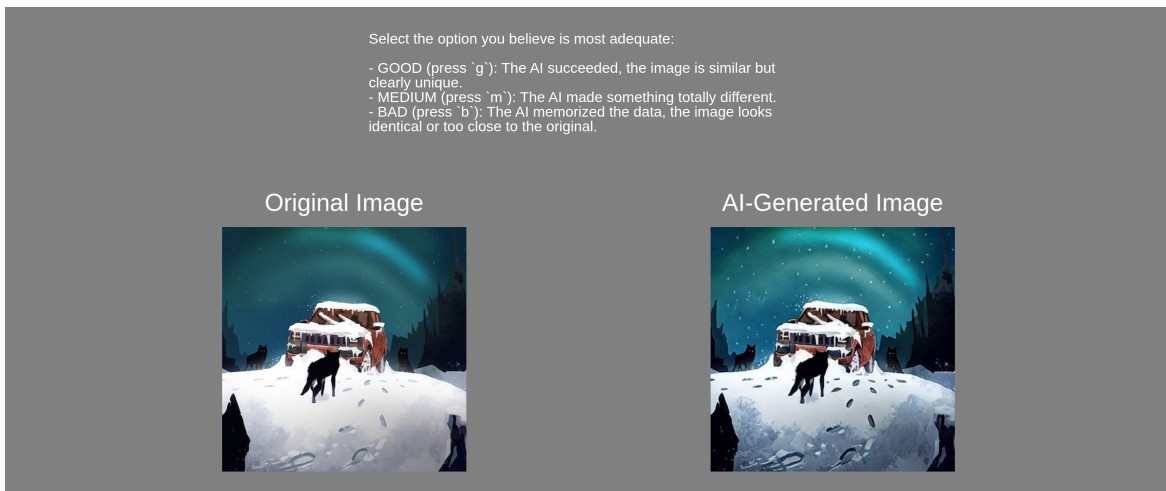

*Figure 8.* Example of a screen displayed to human evaluators, corresponding to a "bad" example.

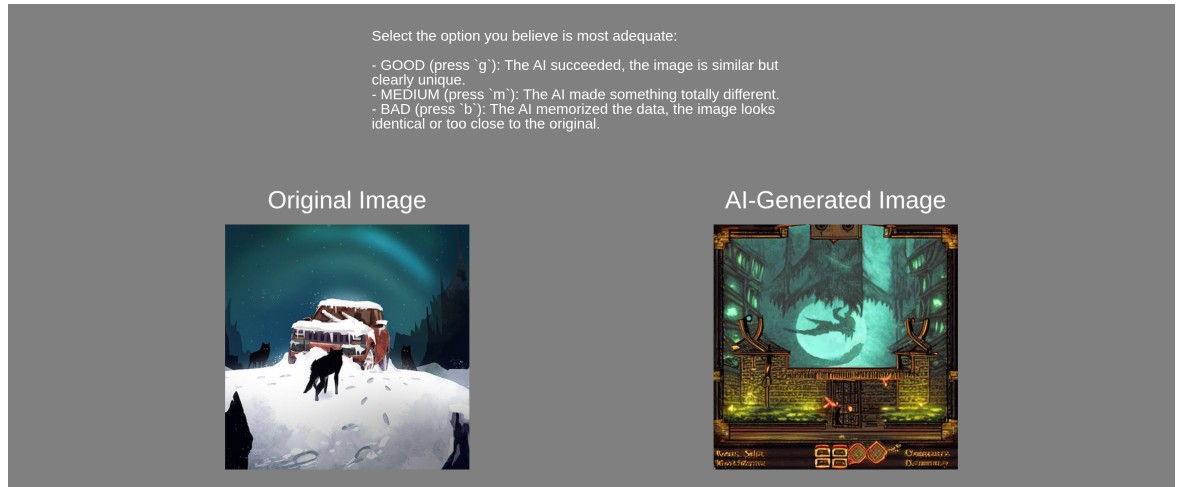

*Figure 9.* Example of a screen displayed to human evaluators, corresponding to a "medium" example.

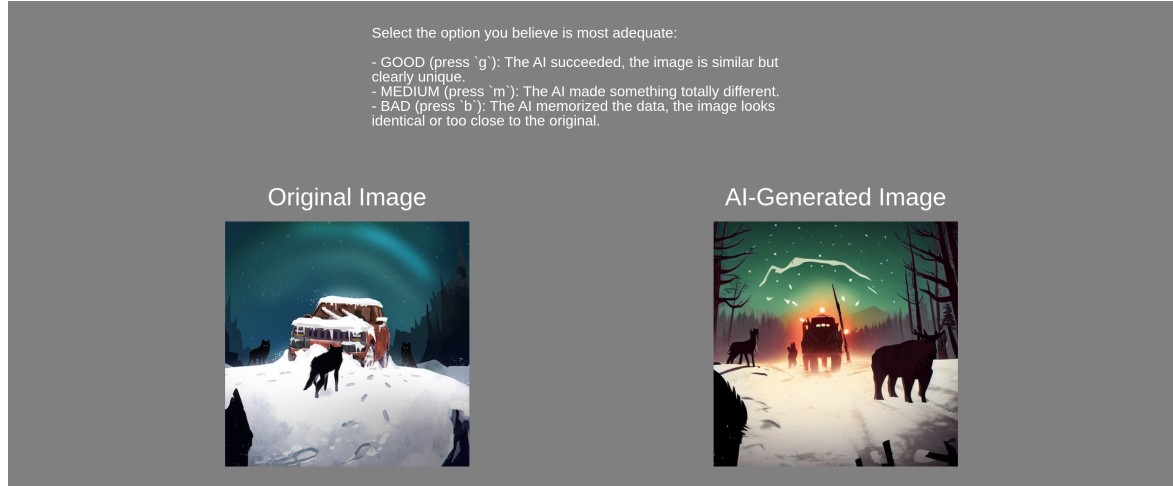

*Figure 10.* Example of a screen displayed to human evaluators, corresponding to a "good" example.

### E.3. Conversational Conversational AI Chatbot

**Dataset Specification.** We utilize the multi-turn human-generated conversations from the ClariQ dataset for our experiments. It contains $498$ human-labeled conversations containing initial questions and further clarifications. Certain conversations include responses which provide little information (e.g., simple responses such as *"No"*, *"I don't know"*, *"I am not sure"*, etc.). To ensure the quality of the data, we filtered out conversations including such responses, resulting in a high-quality dataset of 316 conversations. We use a $50-50$ split for the calibration and test sets.

**Dataset Preprocessing: Question Consolidation.** Each conversation in the dataset is obtained as follows. First, a user asks an ambiguous question, $X_1$. An LLM then generates a follow-up question $Y_1'$ asking for clarification at which point the user provides an answer $X_2'$ to $Y_1'$. The LLM then asks another follow-up question, $Y_2'$, and the user provides an answer $X_3'$. This process happens once more to produce $Y_3'$ and $X_4'$, and results in the sequences $\mathbf{Y}' = (Y_1', Y_2', Y_3')$ and $\mathbf{X}' = (X_2', X_3', X_4')$. The ClariQ dataset contains $X_1$, $\mathbf{Y}'$, and $\mathbf{X}'$; and all the conversations in the dataset are such that at the end of the conversation, the original question has been disambiguated. We then consolidate the conversation into a series of progressively disambiguated questions:

$$X_t = \text{CONSOLIDATE}(X_1, \mathbf{X}'_{:t}, \mathbf{Y}'_{:t-1}) \tag{41}$$

for $t = 2, 3, 4$, where CONSOLIDATE is an LLM call where the LLM is instructed to distill the original question and all provided clarifications into a single question. As a result, the consolidated question $X_t$ represents the entire context of the conversation up to step $t$. We also make another LLM call to obtain an answer $Y_t$ to the consolidated question $X_t$, and the score $S_t^\uparrow = S^\uparrow(X_t, Y_t)$ is obtained through another LLM call where the LLM quantifies how clear $X_t$ is as a question (although the score function could only take $X_t$ as input, we also pass $Y_t$).

Furthermore, to diversify the question sequence lengths over the conversation dataset, we evenly divide the ClariQ dataset (the multi-turn human-generated portion) into $4$ subsets, such that the length of the question sequences (i.e. $\mathbf{X}$) ranges from $1$ to $4$ for each of the subsets. We refer to these subsets as *0-turn questions*, *1-turn questions*, *2-turn questions*, and *3-turn questions*, respectively. To obtain these subsets, for each 4-question sequence we obtained from the consolidation stage, we drop the questions in the middle, such that the initial ambiguous question $X_1$ and the final fully-clarified question $X_4$ are always maintained (with the exception for the *0-turn questions*, for which we only keep $X_4$). As a result, the number of follow-up questions required to obtain an unambiguous question across the test dataset is, on average, *2.5* per conversation. For these conversations, the admissibility $A_t'$ at the $t$-th turn of the conversation is an indicator of whether the original question has been fully disambiguated, i.e., it is $0$ unless $t$ is the final turn of the conversation. Thus, the conformal guarantee ensures that the conversational chatbot stops asking clarifying questions past the turn where the question has been fully disambiguated with probability at least $\gamma$.

**Discussion.** Recall that the selection function is given by the fourth row of Table 1, which does not always automatically satisfy the monotonicity requirement in Definition 1. Note that for this task, $A_t'$ is always non-decreasing in $t$ by construction. As a consequence, $\lambda \mapsto A(\lambda)$ is non-decreasing, and Conf-Gen can thus be applied.

**LLM Prompts for Consolidation, Generation, and Scoring.** We provide in Figure 11 the system prompt we used for consolidation step. The LLM prompt for the response generation given a user question is provided in Figure 12. Lastly, the LLM prompt for the scoring function is provided in Figure 13 and Figure 14. We use *GPT-4o* for the LLM models for question consolidation, response generation, and scoring.

```
consolidation_prompt: |
    You condense a multi-turn passage (alternating lines starting with "User:" and "Assistant:") into a single,
    fully specified question that reflects the user's final clarified intent.
    Rules:
    - Integrate all details; prioritize the latest user clarifications.
    - Resolve pronouns from earlier context when obvious (e.g., \his" → \Michael Jordan").
    - Output ONLY the consolidated question as one sentence, no preamble or extra text.
    - Be concise (<= 25 words) and specific.

consolidation_examples: |
    Example Passages:
    Passage:
    User: How do I bake a cake?
    Assistant: What type of cake are you interested in?
    User: A chocolate cake with minimal sugar.

    Consolidated Question:
    How do I bake a low-sugar chocolate cake?
    ---

    Passage:
    User: Can you tell me about the latest iPhone?
    Assistant: Do you want specifications, price, or features?
    User: Just the new features and improvements.

    Consolidated Question:
    What are the new features and improvements in the latest iPhone?
    ---

    Passage:
    User: I want to plan a vacation.
    Assistant: Where are you thinking of going?
    User: Somewhere in Europe, ideally with historical landmarks.

    Consolidated Question:
    How can I plan a European vacation focused on historical landmarks?
    ---

    Passage:
    User: Explain machine learning.
    Assistant: Do you want a general overview or examples of applications?
    User: Focus on real-world applications in healthcare.

    Consolidated Question:
    What are real-world applications of machine learning in healthcare?
    ---

    Passage:
    User: I need help with my laptop.
    Assistant: What problem are you experiencing?
    User: It keeps overheating when I run multiple programs.

    Consolidated Question:
    How can I prevent my laptop from overheating when running multiple programs?
    ---

    Passage:
    User: I like Italian food.
    Assistant: Are you looking for recipes or restaurants?
    User: Recipes, preferably easy ones I can make at home.

    Consolidated Question:
    What are easy Italian recipes I can prepare at home?
    ---

    Passage:
    User: I want to learn photography.
    Assistant: Do you mean digital or film photography?
    User: Digital, with tips for beginners on portraits.

    Consolidated Question:
    How can a beginner learn digital portrait photography effectively?
    ---

    Passage:
    User: I'm interested in running a marathon.
    Assistant: Do you need training plans or diet advice?
    User: Both, especially training for a first marathon.

    Consolidated Question:
    What training plan and diet advice are recommended for someone running their first marathon?
```

*Figure 11.* Prompt used for the LLM to consolidate questions in ClariQ.

```
generation_prompt: |
  You are a concise evaluator and answer generator.
  Your task is to decide whether the user's question is clear enough to answer directly, and then respond appropriately.
  ---
  Decision rules (in order):
  1. If the question is truly vague, open-ended, or underspecified, respond exactly:
     "I cannot answer because the question is not clear enough."
  2. If the question is reasonably specific and factual | even if it asks for multiple related details (e.g., biography +
  ↪  contributions, or recipes for a specific dish) | treat it as clear and answer concisely (1-2 sentences).
     Only refuse if the intent or essential details are genuinely ambiguous.
  3. If the question is clear but requires real-time data, private information, or external access, respond exactly:
     "The question is specific and clear, but I cannot answer due to information or access limitations."
  ---
  Output rules:
  - Respond with only the answer text | no commentary, explanations, or code formatting.
  - Keep answers short (1-2 sentences), factual, and strictly on-topic.
  - Consider multi-part factual questions as clear if the meaning is specific and understandable.

generation_examples: |
  **Examples:**

  Q: "Tell me about animals."
  A: "I cannot answer because the question is not clear enough."
  ---

  Q: "What are some good ways to train a dog?"
  A: "Positive reinforcement, consistency, and short daily sessions are effective ways to train a dog."
  ---

  Q: "What are some effective exercises to strengthen core muscles and improve posture for office workers?"
  A: "Planks, bridges, and seated stretches help strengthen core muscles and improve posture for office workers."
  ---

  Q: "Who won the FIFA World Cup in 2023?"
  A: "The question is specific and clear, but I cannot answer due to information or access limitations."
  ---

  Q: "Can you tell me about Leonardo da Vinci?"
  A: "Leonardo da Vinci was a Renaissance artist, inventor, and scientist, known for works like the Mona Lisa and The Last
  ↪  Supper."
  ---

  Q: "What are some quick, healthy recipes for dinner under 30 minutes?"
  A: "Grilled salmon with vegetables or stir-fried chicken with broccoli are quick, healthy dinner options under 30 minutes."
  ---

  Q: "Explain AI."
  A: "I cannot answer because the question is not clear enough."
  ---

  Q: "What is the current price of Bitcoin?"
  A: "The question is specific and clear, but I cannot answer due to information or access limitations."
  ---

  Q: "What did Marie Curie contribute to science, and what is her biography?"
  A: "Marie Curie discovered radium and polonium, conducted pioneering research on radioactivity, and was the first woman to win
  ↪  a Nobel Prize; she was born in 1867 and died in 1934."
  ---

  Q: "Give me travel tips."
  A: "I cannot answer because the question is not clear enough."
  ---

  Q: "What are some recommended beginner-friendly books to learn Python programming?"
  A: "'Automate the Boring Stuff with Python' and 'Python Crash Course' are beginner-friendly books to learn Python programming."
  ---

  Q: "Who is the current president of France?"
  A: "The question is specific and clear, but I cannot answer due to information or access limitations."
```

*Figure 12.* Prompt used for the LLM to generate responses to consolidated questions from ClariQ.

```
scoring_prompt: |
  You are evaluating a question-answer pair produced by a separate generation step.
  The question has already been consolidated into a single, clear form.

  Your primary goal is to assign a **continuous clarity and specificity score between 0.0 and 1.0** that reflects
  **how specific, detailed, and well-defined** the question is.

  ### Scoring Philosophy:
  - The score must vary smoothly and **not be limited to round multiples of 0.1.**
  - Use the full numerical continuum | fine-grained differences (e.g., 0.33 vs. 0.37) are encouraged.
  - Imagine the score as a "clarity percentage": 0.00 = completely unclear, 1.00 = perfectly precise.

  ### Main Criteria:
  1. **Clarity & Specificity:**
      - Questions with more explicit context, constraints, or detail score higher.
      - Ambiguous or open-ended questions score lower.

  2. **Answer as Evidence:**
      - The answer serves only as indirect evidence of how clear the question is.
      - A clear question tends to produce a short, precise answer.
         A vague question tends to produce refusal or generalities.
      - However, never inflate the score if the question itself is vague.

  3. **Refusals & Access Limits:**
      - Refusal because of *vagueness* → low score.
      - Refusal because of *access limitation* → high score if question itself is specific.

  ### Interpretation Range:
  | Range | Meaning |
  |--------|----------|
  | 0.00-0.20 | Extremely vague or incoherent |
  | 0.21-0.40 | Somewhat vague, lacks focus |
  | 0.41-0.60 | Moderately clear but missing key details |
  | 0.61-0.80 | Mostly clear and specific, minor ambiguity |
  | 0.81-1.00 | Very clear, precise, and narrowly scoped |

  ### Output format (strict):
  Respond with a single JSON object containing exactly two keys:
  - "score": a continuous float between 0.0 and 1.0 (number, not string).
  - "rationale": one concise sentence explaining why the score was chosen.

  **Rules:**
  - Use precise decimal values (e.g., 0.37, 0.64, 0.82) | avoid rounding to tenths.
  - Output only the JSON object (no code fences, labels, or commentary).
  - Strings inside the JSON must escape all internal quotes (\\").
  - The score must reflect the question's clarity, not how "good" the answer is.
```

*Figure 13.* Prompt used for the LLM to generate responses to consolidated questions from ClariQ.

```
scoring_examples: |
  **Examples:**
  # ====== Very Low Clarity (0.0-0.2) ======
  Q: "Tell me something interesting."
  A: "I cannot answer because the question is not clear enough."
  Score: 0.05
  Rationale: No topic or constraint; completely open-ended and vague.
  Q: "Explain."
  A: "I cannot answer because the question is not clear enough."
  Score: 0.02
  Rationale: Incoherent and contextless; provides no subject to address.
  Q: "Talk about AI."
  A: "I cannot answer because the question is not clear enough."
  Score: 0.12
  Rationale: Mentions a topic but lacks any direction, focus, or goal.
  Q: "Tell me about history."
  A: "I cannot answer because the question is not clear enough."
  Score: 0.15
  Rationale: Very broad; no timeframe, region, or event specified.
  Q: "How are things?"
  A: "I cannot answer because the question is not clear enough."
  Score: 0.08
  Rationale: Conversational but meaningless as a factual query.
  # ====== Low-Moderate Clarity (0.21-0.4) ======
  Q: "What are some technologies used today?"
  A: "Computers, smartphones, and the internet are common technologies today."
  Score: 0.27
  Rationale: Some topical relevance but extremely broad; lacks domain or purpose.
  Q: "Tell me about AI in healthcare."
  A: "AI in healthcare includes diagnosis support and predictive analytics."
  Score: 0.34
  Rationale: Slightly more focused but still too general and undefined in scope.
  Q: "What are ways to improve health?"
  A: "Exercise, balanced diet, and adequate sleep improve health."
  Score: 0.31
  Rationale: Basic but non-specific; does not define population or goal.
  Q: "Why is the sky blue?"
  A: "Because of Rayleigh scattering of sunlight in the atmosphere."
  Score: 0.39
  Rationale: Simple but specific physical question; focused yet shallow in scope.
  # ====== Moderate Clarity (0.41-0.6) ======
  Q: "What are some ways to reduce carbon emissions?"
  A: "Renewable energy, energy efficiency, and carbon taxes can reduce emissions."
  Score: 0.57
  Rationale: Moderately specific; identifies a theme but lacks timeframe or region.
  Q: "How does machine learning work?"
  A: "It learns patterns from data to make predictions or decisions."
  Score: 0.45
  Rationale: Clear topic but broad and missing key context or examples.
  Q: "What are the causes of climate change?"
  A: "Main causes include burning fossil fuels, deforestation, and agriculture."
  Score: 0.53
  Rationale: Understandable but still general; no particular framing or focus.
  Q: "What are benefits of meditation?"
  A: "It reduces stress and improves focus and emotional well-being."
  Score: 0.49
  Rationale: Clear but not specific; lacks context such as duration or demographic.
  Q: "Explain the difference between a virus and bacteria."
  A: "Viruses need a host to reproduce, while bacteria can live independently."
  Score: 0.58
  Rationale: Narrow topic but not framed in detail (e.g., structure, examples).
  # ====== Fair-High Clarity (0.61-0.8) ======
  Q: "What are common side effects of taking antibiotics?"
  A: "Nausea, diarrhea, and allergic reactions are common side effects."
  Score: 0.69
  Rationale: Fairly clear and focused but lacks mention of drug type or duration.
  Q: "How can students improve concentration while studying?"
  A: "Taking breaks, avoiding distractions, and maintaining good sleep can help."
  Score: 0.64
  Rationale: Clear practical goal but lacks specificity (e.g., age group or context).
  Q: "What are the main causes of inflation in developed economies?"
  A: "Supply chain issues, monetary policy, and demand surges are main causes."
  Score: 0.77
  Rationale: Clear economic scope and focus; moderately well-framed.
  Q: "What are key differences between renewable and non-renewable energy sources?"
  A: "Renewables are replenishable (solar, wind), while non-renewables are finite (coal, oil)."
  Score: 0.73
  Rationale: Well-scoped and comparative but lacks application context.
  Q: "What are early symptoms of diabetes in adults?"
  A: "Frequent urination, increased thirst, and fatigue are early symptoms."
  Score: 0.81
  Rationale: Highly focused medical question with clear population definition.
```

*Figure 14.* Prompt used for the LLM to score questions from ClariQ.

```
# ====== High Clarity (0.81-0.95) ======
  Q: "What are the top three renewable energy strategies for reducing carbon emissions in urban areas by 2030?"
  A: "Solar rooftops, electric public transport, and building efficiency improvements."
  Score: 0.92
  Rationale: Very specific | defined goal, context, timeframe, and scope.
  Q: "How can a beginner learn Python for data analysis within three months?"
  A: "Take structured courses, practice datasets, and complete small projects."
  Score: 0.87
  Rationale: Clear, focused, and practical; specifies skill level, purpose, and timeframe.
  Q: "What exercises are recommended for improving cardiovascular health in adults over 50?"
  A: "Brisk walking, cycling, and swimming are recommended."
  Score: 0.93
  Rationale: Clear, medically targeted, and demographically scoped.
  Q: "What are the differences between supervised and unsupervised learning, with examples for each?"
  A: "Supervised learning uses labeled data (classification); unsupervised finds clusters."
  Score: 0.96
  Rationale: Exceptionally precise; question explicitly defines scope and deliverables.
  Q: "What were the economic effects of the 2008 financial crisis on European employment between 2009 and 2012?"
  A: "Unemployment rates rose significantly, especially among young workers."
  Score: 0.94
  Rationale: Detailed timeframe, region, and economic focus yield very high clarity.
  # ====== Perfect Clarity (1.0) ======
  Q: "What is the acceleration due to gravity on Earth at sea level, in meters per second squared?"
  A: "Approximately 9.81 m/s²."
  Score: 1.0
  Rationale: Fully specific, measurable, and unambiguous; perfect clarity.
  Q: "In what year did the Apollo 11 mission land on the Moon?"
  A: "1969."
  Score: 0.99
  Rationale: Exact factual target, zero ambiguity.
  Q: "Which programming language introduced the concept of object-oriented programming first?"
  A: "Simula, introduced in the 1960s, was the first object-oriented language."
  Score: 0.98
  Rationale: Clear, historical, and narrowly defined.
  Q: "What is the melting point of pure gold in degrees Celsius?"
  A: "1064°C."
  Score: 0.99
  Rationale: Precise scientific measurement with no ambiguity.
  # ====== Calibration (fine granularity guidance) ======
  The following examples illustrate subtle scoring differences between similar questions, to encourage
  non-round, continuous scoring behavior.
  # ====== Fine-grained calibration examples ======
  Q: "Tell me about renewable energy."
  A: "I cannot answer because the question is not clear enough."
  Score: 0.25
  Rationale: Broad topic without context or goal; only slightly better than incoherent because it names a concept.
  Q: "What are types of renewable energy sources?"
  A: "Solar, wind, hydro, and geothermal are main renewable energy types."
  Score: 0.37
  Rationale: Slightly clearer; enumerative but still lacks focus, timeframe, or purpose.
  Q: "What are the main advantages of renewable energy?"
  A: "It's sustainable, reduces emissions, and lowers long-term costs."
  Score: 0.44
  Rationale: Defines direction (\advantages"), improving specificity, though still broad in scope.
  Q: "What are the main advantages of solar energy in residential use?"
  A: "It reduces electricity bills and reliance on the grid."
  Score: 0.68
  Rationale: Sharply more focused; defines technology (solar) and context (residential).
  Q: "What are the main advantages of rooftop solar systems in reducing household electricity costs in Canada?"
  A: "They provide stable long-term savings and reduce grid dependency."
  Score: 0.79
  Rationale: Very specific geographic and functional framing; near high clarity but still not numerical or time-bounded.
  Q: "What is the estimated average payback period, in years, for a 5kW rooftop solar system in Ontario under current energy
  ↪  prices?"
  A: "Around 8 to 10 years, depending on usage and incentives."
  Score: 0.93
  Rationale: Fully concrete parameters (location, system size, metric) make this almost perfectly precise.
```

*Figure 15.* Prompt used for the LLM to score questions from ClariQ (continued).

### E.4. Conformal Agentic AI for Web-Based Tasks

**Dataset and Task Setup.** We utilize the benchmark dataset proposed by He et al. (2024), which initially comprises 643 tasks spanning 15 diverse websites, including Amazon, BBC News, Google Flights, and Wolfram Alpha. To ensure a reliable evaluation environment, we filter out tasks rendered infeasible by external factors such as unremovable ads, mandatory login requests, or strict anti-bot policies. This filtering process results in a curated set of 542 tasks. We partition these tasks into a calibration set (80%) and a test set (20%).

**Sampling Strategy.** For each valid task in our dataset, we generate up to 10 candidate action trajectories using an iterative agentic framework. More specifically, we always attempt to generate 10 trajectories, but due to occasional API call errors, some tasks end up with 8 or 9 trajectories. To generate an action trajectory, we employ GPT-4.1 in an iterative process. At each step, it is presented with a screenshot of the current browser view and the task context. The model then samples a specific UI action (e.g., clicking a coordinate, scrolling, or typing). This action is executed by the backend browser, updating the state for the next step. Detailed specifications of the action space and browser environment follow those from He et al. (2024).

**Score Function.** To obtain the score function, we require the LLM to self-assess its progress at every step of the action trajectory. The model outputs a floating-point number in the range $[0, 1]$, quantifying the percentage of the task it believes has been completed (with 1 indicating full success). We achieve this by appending a specific scoring instruction to the original prompt template from He et al. (2024). The instruction is `Completion: {A float between 0 and 1 indicating how much of the task has been completed until now. Zero means no progress has been made and one means full completion.}`. Placing this instruction at the end allows us to extract the score while preserving the integrity of the original generation context. Ultimately, we define the score of a full trajectory as the self-assessment value provided at its final step.

**Admissibility Function.** To implement our admissibility function, we adopt the automated evaluation protocol established by He et al. (2024). We employ GPT-4.1 as an automated judge, denoted by $A'$, to evaluate the performance of action trajectories. Specifically, for a task $x$ and a trajectory of actions $\mathbf{y}_t$, the function $A'(x, \mathbf{y}_t)$ assigns a binary label of 1 (success) or 0 (failure) to $\mathbf{y}_t$ by analyzing the task description alongside the complete history of screenshots and text outputs; note that there is no ground truth here. We adhere strictly to the assessment prompts and criteria detailed in the original study.

### E.5. Conformal Random Forests

**Detailed Method.** We trained $T$ trees $F_1, \ldots, F_T$ using a training dataset. Here $F_t : \mathcal{X} \to \triangle^{C-1}$, i.e., given a feature vector $X$, each tree will output a probability vector over $C$ possible classes. The prediction of the random forest model $F$ will be $F(X) = \frac{1}{T} \sum_{t=1}^{T} F_t(X)$.

In our setting, given a tree $F_t$, one can retrieve the prediction $F_t(X)$. As our scoring function should reflect confidence in the correctness of $F_t(X)$, we use the weighted number of samples of the predicted leaf of $X$ in the tree $F_t$ as a proxy. This can be achieved by using the `weighted_n_node_samples` in the `Tree` class in the `sklearn` Python package (Pedregosa et al., 2011). Let $W_t(X)$ be the weighted number of samples of the predicted leaf of $X$ in the tree $F_t$. Then we can let $Y_t = (F_t(X), W_t(X))$. Then the score function $S^{\uparrow}(X, Y_t) = W_t(X)$. We used the `SmallestSubsetSum` selection function described in Appendix D.

The admissibility $A'_t \in \{0, 1\}$ is 1 if and only if $F_t$ predicted the correct class, i.e., $\arg\max F_t(X) = Y_{\text{GT}}$.[8] The `agg` function is 1 if the number of trees predicting the correct class is at least $k$, where $k$ is a hyperparameter, and 0 otherwise. This choice corresponds to Equation 37 with $\beta = k$ (except $A'_t$ is different in this experiment than in the example in Appendix D). Thus the conformal guarantee is that at least $k$ trees predicted the correct class.

The dataset Click_prediction_small contains 35,953 train instances and 3,995 test instances, with 11 features and 2 classes. This data is derived from the 2012 KDD Cup. The data is subsampled to 0.1% of the original number of instances, downsampling the majority class (click=0) so that the target feature is reasonably balanced (5 to 1). The data is about advertisements shown alongside search results in a search engine, and whether or not people clicked on these ads. The task is to build the best possible model to predict whether a user will click on a given ad.

---

[8]Note that this is an example where the elements of $\mathbf{Y}$ do not live in the same space as $Y_{\text{GT}}$: the former are pairs $(F_t(X), W_t(X))$, and the later is a class label. In other words, this example highlights that Conf-Gen remains valid when $\mathcal{Y} \neq \mathcal{Y}_{\text{GT}}$.

In addition to the dataset Click_prediction_small, we performed experiments on four more datasets; we show the results below.

**Dataset GesturePhaseProcessed.** The dataset GesturePhaseProcessed (Madeo et al., 2013) contains 8,885 train instances and 988 test instances, with 32 features and 5 classes. The dataset is composed of features extracted from 7 videos with people gesticulating, and aims at studying gesture phase segmentation. Each video is represented by two files: a raw file, which contains the position of hands, wrists, head, and spine of the user in each frame; and a processed file, which contains velocity and acceleration of hands and wrists. We trained the random forest model on the train instances to achieve an AUC of 0.9027. Figure 16 illustrates the average admissibility of the selected conformal sets on GesturePhaseProcessed. The diagonal line represents the theoretical lower bound established in Theorem 3, which is empirically satisfied for all values of $\gamma \in [0, 1]$. Figure 16 (right) plots the average number of trees used as a function of $\gamma$.

**Dataset Adult.** The dataset Adult (Becker & Kohavi, 1996) contains 43,957 train instances and 4,885 test instances, with 14 features and 2 classes. The dataset prediction task is to determine whether a person makes over 50K a year. Extraction was done by Barry Becker from the 1994 Census database. We trained the random forest model on the train instances to achieve an AUC of 0.9104. Figure 17 illustrates the average admissibility of the selected conformal sets on Adult. The diagonal line represents the theoretical lower bound established in Theorem 3, which is empirically satisfied for all values of $\gamma \in [0, 1]$. Figure 17 (right) plots the average number of trees used as a function of $\gamma$.

**Dataset Census-Income.** The dataset Census-Income (U.S. Census Bureau, 2000) contains 269,356 train instances and 29,929 test instances, with 41 features and 2 classes. This dataset contains weighted census data extracted from the 1994 and 1995 "Current Population Surveys" conducted by the U.S. Census Bureau. The data contains 41 demographic and employment related variables. We trained the random forest model on the train instances to achieve an AUC of 0.9478. Figure 18 illustrates the average admissibility of the selected conformal sets on Census-Income. The diagonal line represents the theoretical lower bound established in Theorem 3, which is empirically satisfied for all values of $\gamma \in [0, 1]$. Figure 18 (right) plots the average number of trees used as a function of $\gamma$.

**Dataset MiniBooNE.** The dataset MiniBooNE (Roe, 2005) contains 117,057 train instances and 13,007 test instances, with 50 features and 2 classes. This dataset is taken from the MiniBooNE experiment and is used to distinguish electron neutrinos (signal) from muon neutrinos (background). We trained the random forest model on the train instances to achieve an AUC of 0.9790. Figure 19 illustrates the average admissibility of the selected conformal sets on MiniBooNE. The diagonal line represents the theoretical lower bound established in Theorem 3, which is empirically satisfied for all values of $\gamma \in [0, 1]$. Figure 19 (right) plots the average number of trees used as a function of $\gamma$.

**Difference in performance.** We perform experiments on varying the strength of the conformal guarantee $k$, and note their difference in the performance of the model by selecting the best subset of the 100 trees. To this end, for each $k$ from 5 to 50, with an increment of 5, and for each $\gamma$, we perform calibration on 100 instances in the test set. Then, we perform conformal generation on the remaining instances in the test dataset and compute the performance using AUC with respect to the ground truth label. For each $k$, we pick the $\gamma$ with the highest AUC. Figure 20 shows the change in AUC for each $k$. We note that for most datasets, if $k$ is in a reasonable range, the change in AUC is minimal. For Click_prediction_small, there is a minor increase in the AUC using this method. This shows that it is possible to gain AUC "for free" by using an existing model and just a few data instances on the calibration dataset.

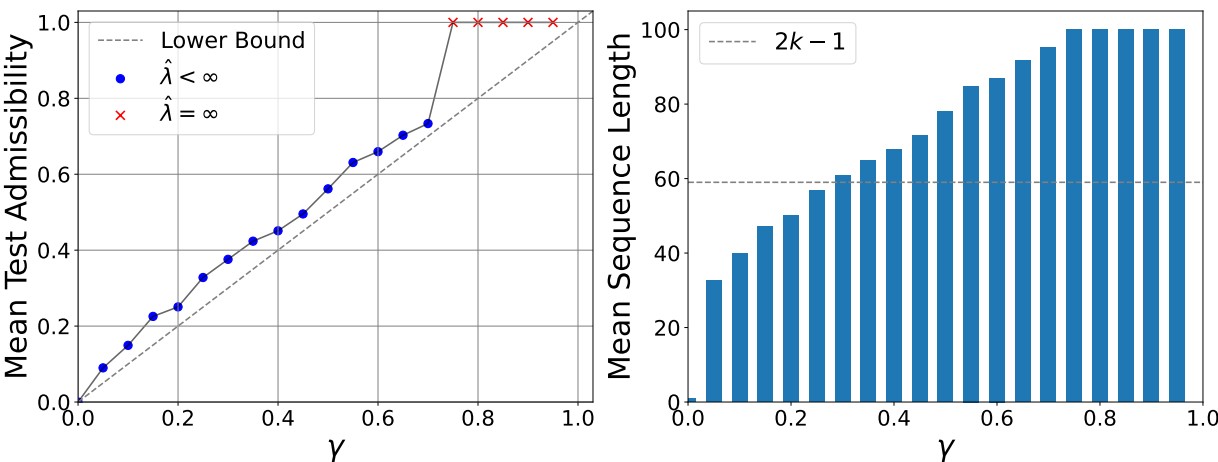

*Figure 16.* Conformal generation admissibility and average sequence length as a function of $\gamma$ of our conformal random forest on GesturePhaseProcessed.

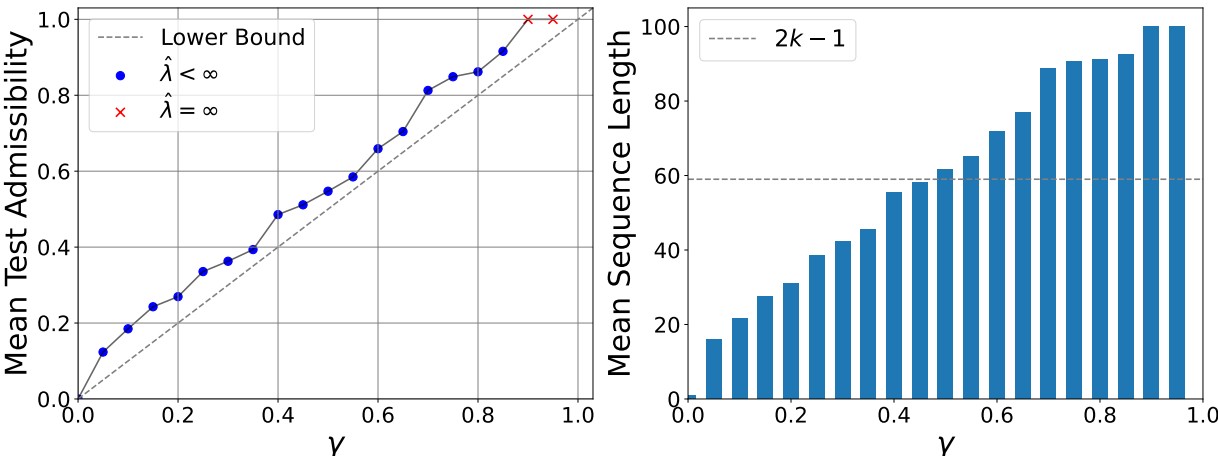

*Figure 17.* Conformal generation admissibility and average sequence length as a function of $\gamma$ of our conformal random forest on Adult.

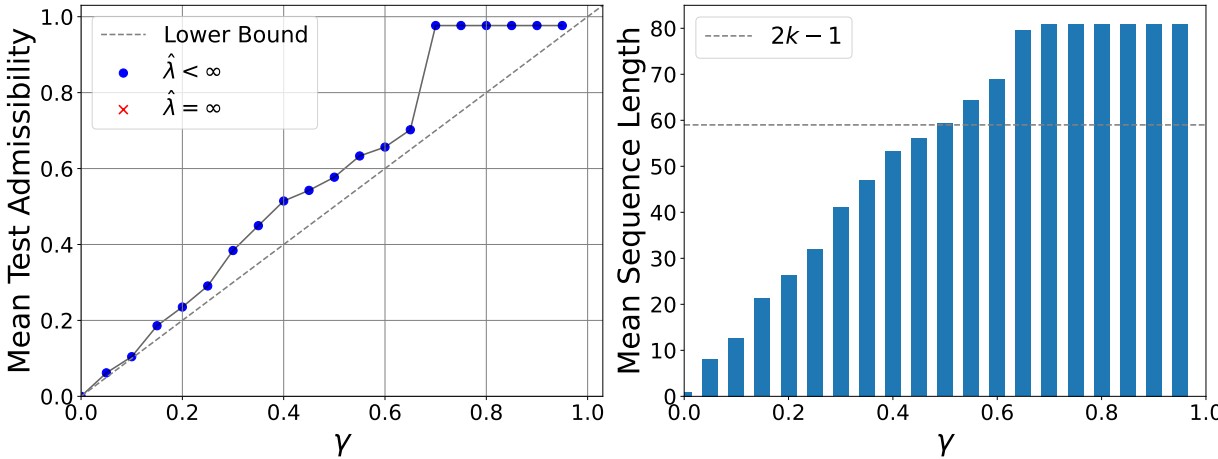

*Figure 18.* Conformal generation admissibility and average sequence length as a function of $\gamma$ of our conformal random forest on Census-Income.

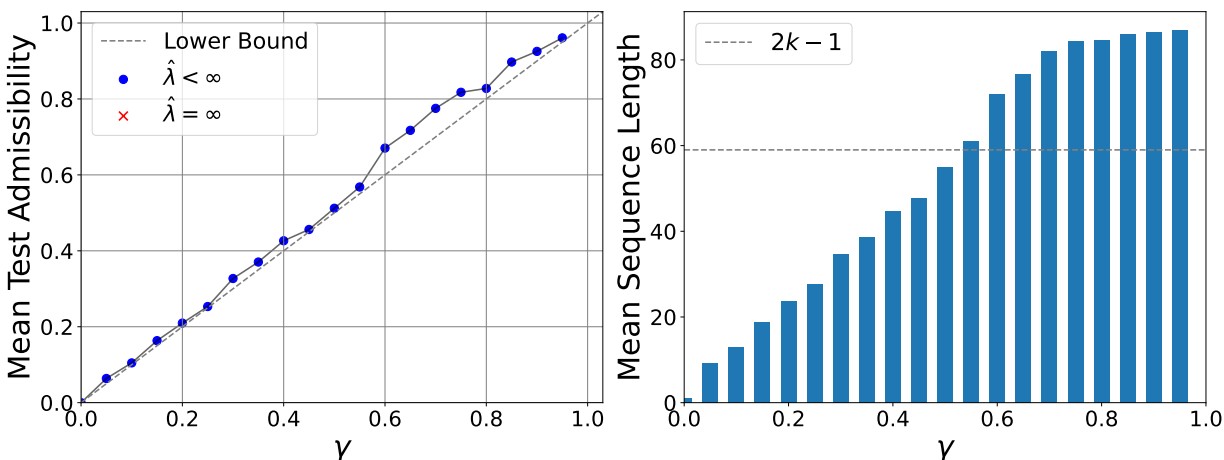

*Figure 19.* Conformal generation admissibility and average sequence length as a function of $\gamma$ of our conformal random forest on MiniBooNE.

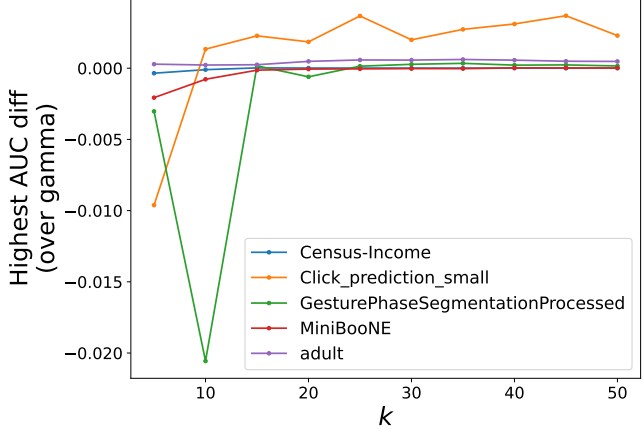

*Figure 20.* Change in performance (AUC) of our conformal random forests for various datasets and $k$ values.

