# OpenReview forum: "Conf-Gen: Conformal Uncertainty Quantification for Generative Models"
_ICML.cc/2026/Conference — ICML 2026 regular_

### Official Review · Reviewer_2YY2 · 2026-02-24

**Soundness:** 2
**Presentation:** 3
**Significance:** 2
**Originality:** 2
**Overall Recommendation:** 3
**Confidence:** 3

**Summary:**

This paper present an extension of conformal prediction to generative models.

**Compliance With Llm Reviewing Policy:**

Affirmed.

**Final Justification:**

I do not find that author's rebuttal clearly address my concerns, maybe I am wrong but I still suspect that the main claimed theoretical contribution is not very significant. And also, experiments are quite confusing. I keep my original rating and confidence.

**Key Questions For Authors:**

- Could you state explicitly why Conf-Gen is not equivalent to CRC? Maybe I'm wrong but I feel like Conf-Gen can be seen as a particular case of CRC.
- Several times in the paper you claim that your method does not necessitate ground truth to be usable, which make it different from CRC. But, in CRC, if $U$ does not depend on $Y_{gt}$, isn't it the same?
- Could you give some interpretation of the lower bound (9) in Thm 3?

**Limitations:**

Yes

**Strengths And Weaknesses:**

## Strength
**Soundness**
- The paper proposes a clear theoretical approach of the problem
- Their extension of CRC is justified theoretically by theorem 3.
- Experiments over several domains (text and images generation)

**Presentation**
- Presentation is easy to follow
- Writing is clear

**Significance**
- The problem of uncertainties in generative models is a very important and interesting one

**Originality**
- Experimental results on tasks where conformal predictions have not been applied before
- Theorem 3 provides a new extension to CRC.

## Weakness
**Soundness**
- I am wondering how this paper really differs from original CRC, see questions
- I would have also expected greater discussions on the limitations of the actual conformal prediction framework, like dependence on the calibration set, variance, etc. I find that the fact that all expectations are over the calibration set is not emphasized enough and neither reflected in the experimental parts.

**Presentation**
- Related to above, I find that there are too many defensive claims on how this paper is new and differs from CRC
- While there are some details in the appendix, I find that the main parts could benefit from much more experimental details. I find it hard here to even guess what kind of evaluation was conducted.
- Still regarding the experiments, more comprehensive experimental details or reporting on the score, the admissibility functions or $C_\lambda$ would greatly help the reader to appreciate what has been done. Maybe a table explaining clearly for each experiments how the generation is conducted, which score is used, etc could help.
- Overall the experimental parts feels a bit weak, I would have expected greater concrete discussions around what Conf-Gen concretely brings. This could help make this work more convincing.
- Part. 4 on computational considerations lacks concrete grounding, and, in my opinion, not very relevant in the narrative flow of the paper.
- I find some notations and precision to be too verbose and confusing like footnote 1 l.105 and some paragraphs, or the redundant precision that some functions may be stochastic.
- Overall, I think that some parts could be streamlined for a better global comprehension and more technical details to be explained cleanly in the appendix.

**Originality**
Several aspects have already been mentioned above, but overall:
- This paper seems to reformulate several existing approaches rather than proposing something new, see questions.

---

> ### Author Rebuttal · Authors · 2026-03-27
>
> Thank you for reviewing our work. We are glad you found that the problem we study is “very important and interesting” and that our “writing is clear”. Below we address the issues you raised in your review; please let us know if you have any lingering concerns. We believe the discussion below addresses your concerns, if you agree, we ask that you please consider raising your score. We will clarify the points below in our camera-ready version if our paper is accepted.
>
> > On the differences between Conf-Gen and CRC.
>
> There’s two axes along which Conf-Gen is different from CRC. The first one is about how CRC is used in practice: $x$ corresponds to “features” or to the “input” of a supervised ML model, the ML model is used to construct a score function, which in turn yields $C_\lambda(x)$—which is always a set— $y_{GT}$ is a ground truth label, and the function $U(C, y_{GT})$ must be callable to perform calibration. Although you are correct that, mathematically we could ignore the dependence of $U$ on $y_{GT}$, this would involve $U$ corresponding to a correctness oracle (e.g. human evaluation), which is not callable. Also, in CRC there is no notion of “generations” and users have no control over the distribution of the “features”, as they are dictated by the task. In contrast, in Conf-Gen we do not require $A(x, C, y_{GT})$ to be callable, and instead rely just on a calibration dataset—this is what allows us to ignore $y_{GT}$ when needed and still perform calibration. Additionally, the ML model is not only used within the score function, but it also produces generations $\textbf{y}$; this means that what corresponds to the “features”, $(x, \textbf{y})$, now contains both the input and output of the generative model, and so the user has some control over their distribution. Additionally, $C_\lambda$ need not be a set. We highlight that **although these differences are practical and not mathematical, they are crucial: they are precisely what enables all of our novel applications**.
>
> The second perspective is purely mathematical: CRC assumes some regularity conditions on the functions $\lambda \mapsto U^{(i)}(C_\lambda(X^{(i)}), Y_{GT}^{(i)})$, and analogously, we require regularity conditions on the functions $\lambda \mapsto A(X^{(i)}, C_\lambda(X^{(i)}, \textbf{Y}^{(i)}), Y_{GT}^{(i)})$. In this perspective, how these functions differ from each other is not relevant, rather we just focus on the regularity conditions and the resulting guarantees. The resulting guarantees are indeed the same, but **the regularity conditions of Conf-Gen are strictly weaker than those of CRC**. Several of these weaker conditions are minor technical improvements, but the weakening of the monotonicity assumption is particularly relevant, e.g. it enables the task of generating non-memorized images. We included an in-depth discussion of these technical differences in Appendix C.
>
> We will better delineate between practical and mathematical differences between CRC and Conf-Gen when updating the paper.
>
> > On the upper bound (eq 9):
>
> The supremum in the definition of H (eq 10) is the largest value of $\overline{A}\_{n+1}$ before $\lambda^{\ast\ast}$. Since $\lambda^{\ast\ast}$ is defined as the first $\lambda$ for which $\overline{A}\_{n+1}$ exceeds $\gamma + a\_{max}/(n+1)$, this means that $H$ must be nonnegative and that it quantifies the smallest gap between $\overline{A}\_{n+1}(\lambda^{\ast\ast})$ and $\overline{A}\_{n+1}$ at smaller values of $\lambda$. When $\mathbb{E}[H]$ is small, the upper bound closely tracks the lower bound; e.g. if $\overline{A}\_{n+1}$ is continuous, then $H=0$, resulting in the tightest possible gap between bounds. On the other hand, if $\overline{A}\_{n+1}$ has large jumps, $\mathbb{E}[H]$ will be larger. Note that we included a discussion of the upper bound in Appendix C.2; we will enhance it with this discussion.
>
> > On presentation, experimental details, and discussing limitations:
>
> We will incorporate this feedback into our camera-ready version. We do nonetheless highlight that the limitations you brought up are common to all conformal methods, and that Appendix E already includes the requested experimental details (although we will highlight them more clearly to avoid them being hidden in text).

---

> > ### Author Rebuttal · Reviewer_2YY2 · 2026-04-02
> >
> > I thank the authors for their answers.
> > However I consider their answers to not be precise enough to solve my concerns. I think this answer is more confusing than a clear and clean mathematical answer to my questions.
> > Also, for the upper bounds I did not asked for a textual re-framing of the results but more for a "intuitive" interpretation.
> >
> > I will maintain my score and confidence.

---

> > > ### Author Response · Authors · 2026-04-06
> > >
> > > Thank you for your continued engagement!
> > >
> > > Can you please clarify in what sense our response is not clear enough? You mention we did not provide a “clean mathematical answer” to your questions; this is because we did not interpret your questions as requiring such an answer, but rather an intuitive one. The clean mathematical answer to how Conf-Gen differs from CRC lies in the difference in assumptions of Theorems 2, 3, and 5; we discussed these differences in the paper and included a more detailed discussion in Appendix C. The assumptions are stated in unambiguous and mathematically formal terms in the paper, which is why we focused our answer on the intuitive differences. We also highlight that, if the concern is lack of novelty over CRC, we presented 4 novel tasks to which conformal methods—including CRC— had not been applied before.
> > >
> > > As for the upper bound, $H$ can be informally understood as quantifying the non-infinitesimal changes (jumps) that can happen to the admissibility when infinitesimal changes are made to $\lambda$: the smaller the jumps, the closer to zero $H$ will be, and thus the gap between the lower and upper bounds will also be smaller. This is why when $\bar{A}_{n+1}$ is continuous in $\lambda$, we have that $\mathbb{E}[H] = 0$.

---

### Official Review · Reviewer_NiUg · 2026-03-07

**Soundness:** 4
**Presentation:** 4
**Significance:** 3
**Originality:** 3
**Overall Recommendation:** 5
**Confidence:** 4

**Summary:**

The authors present an extension of conformal risk control (CRC) which is itself an extension of conformal prediction (CP), suited to obtaining performance guarantees for the answers provided by LLMs, in the case where there is a set of exchangeable (input, ground_truth) pairs. One has to also interpret the LLM output to form a set or sequence of candidate answers parametrized by a conservativenss parameter lambda (e.g., to select the most promising candidates) and provide both a way to obtain a score for each candidate answer and a way to evaluate how good this output is given a ground truth answer. A calibration method then allows to choose lambda to obtain a required degree of expected goodness on a new example from the same distribution.

**Compliance With Llm Reviewing Policy:**

Affirmed.

**Final Justification:**

My initial (rather positive) judgement stands.

**Key Questions For Authors:**

One limitation I would note is that the kind of data stream on which LLMs are actually applied is not necessarily exchangeable, invalidating the core assumption behind CP. But the guarantees may be useful in practice nonetheless, e.g., to obtain conservative safety guarantees. Another CP limitation of course is that these are only guarantees in average, not for a particular query. This is of course not a specific issue with this paper, but it may limit the usefulness of this family of approach, e.g., in high-stakes domains like medical decision-making. Please share any thoughts on these two issues (non-exchangeability and average-guarantees).

**Limitations:**

I haven't seen anything discussed on limitations.

**Strengths And Weaknesses:**

Soundness and originality

This paper allows doing a form of calibration with conformal guarantees on LLM outputs in settings not considered before, which makes comparisons difficult in several of these settings.

The experiments suggest that the theoretical guarantees hold.

The paper appears to be very sound and original.

Presentation

The paper does a good job of presenting relevant prior work.

The formal aspect of the paper also appears much above the ICML average and I did not detect any issue on that front, although I did not read the appendix.

More generally, everything is very clear.

Significance

This extension of conformal prediction seems relevant for many LLM-related tasks and could thus be impactful.

See the key questions for authors.

---

> ### Author Rebuttal · Authors · 2026-03-27
>
> Thank you for your positive review. We are glad you found our work “very sound and original” and “relevant for many LLM-related tasks and could thus be impactful”. Below we address the questions you asked in your review; please let us know if you have any lingering concerns. We will clarify the points below in our camera-ready version if our paper is accepted.
>
> > On the LLM data streams not being exchangeable:
>
> While indeed LLM data streams are not exchangeable, **this does not violate our exchangeability requirements nor our conformal guarantees**: we do not assume exchangeability between different tokens of a single response, nor between different responses of a single sequence of responses. We only assume exchangeability between different sequences of generated responses, i.e. the sequences themselves must be exchangeable, not the elements within them.
>
> > On the guarantees only holding on average:
>
> This is indeed the case. As you correctly point out, this is a general property of conformal methods like CP and CRC, not just Conf-Gen. We will make sure to mention this in the final version of the paper.

---

> > ### Author Rebuttal · Reviewer_NiUg · 2026-04-05
> >
> > Thank you for acknowledging my points.

---

### Official Review · Reviewer_fyDF · 2026-03-13

**Soundness:** 3
**Presentation:** 1
**Significance:** 3
**Originality:** 2
**Overall Recommendation:** 4
**Confidence:** 2

**Summary:**

This paper introduces Conformal Generation (Conf-Gen), a framework for applying conformal guarantees to generative tasks where outputs are structured objects (e.g., sequences) and ground truth may be ambiguous. The key idea is to define a conformal procedure over generated outputs using an admissibility function that evaluates generated objects relative to an input and possibly a generation sequence. The framework generalizes conformal risk control and provides theoretical guarantees under a new “$\gamma$-sensibility” condition. The authors derive lower and upper bounds on expected admissibility and discuss computational strategies for efficient calibration (e.g., score-based selectors and partial generation). The framework is demonstrated on several tasks including open-domain QA, image memorization detection, conversational clarification or web agents.

**Compliance With Llm Reviewing Policy:**

Affirmed.

**Ethical Review Concerns:**

The paper includes experiments related to memorization in generative image models and relies heavily on automated LLM evaluators. The ethical implications of these setups, including reliability of proxy judges and potential copyright/privacy considerations in memorization analysis, are not sufficiently discussed.

**Ethical Review Flag:**

Flag this paper for an ethics review.

**Ethics Expertise Needed:**

["Responsible Research Practice (e.g., IRB, documentation, research ethics)", "Privacy and Security (e.g., personally identifiable information)", "Legal Compliance (e.g., EU AI Act, GDPR, copyright, terms of use)"]

**Final Justification:**

The authors' responses addressed some of my concerns, and I think the paper is a meaningful extension of the theory of Conformal Prediction. However, there are still additional issues that need to be improved (e.g. evaluations, remarks by the other reviewers etc.).

**Key Questions For Authors:**

- How should practitioners verify the $\gamma$-sensibility assumptions in practice? Providing diagnostics, sufficient conditions, or empirical validation strategies would strengthen the applicability of the framework.
- Can the authors provide stronger empirical comparisons for at least one of the new application domains (e.g., conversational or agentic settings)? Baselines or ablations would help demonstrate practical benefits beyond feasibility.
- How sensitive are the results to the choice of LLM-based judges used for admissibility or evaluation? Some discussion or robustness analysis would clarify how meaningful the guarantees are under different evaluators.
- For the image memorization experiment, can the authors report annotator agreement and clarify the rationale for the admissibility labeling scheme?

**Limitations:**

The paper briefly mentions potential societal consequences but does not meaningfully discuss limitations. In particular, it should discuss reliance on proxy judges, assumptions required for the guarantees, risks in agentic applications, and implications of memorization detection in generative image models.

**Strengths And Weaknesses:**

### Strengths
- The paper proposes a general framework for conformal guarantees in generative settings, unifying several task-specific approaches under a single abstraction.
- The theoretical contribution is meaningful: the analysis extends conformal risk control and introduces guarantees under a weaker assumption (monotonicity in conditional expectation rather than almost-sure monotonicity).
- The topic is timely and relevant given growing interest in uncertainty quantification for generative models.
- The paper demonstrates the framework across diverse application domains (QA, image generation, conversational agents, web agents), which helps illustrate its flexibility.
---
### Weaknesses
- The main theoretical guarantee relies on the $\gamma$-sensibility assumption, particularly conditional-expectation monotonicity, which may be difficult to verify in real generative systems. In several experiments this assumption is argued informally rather than validated.
- Many experiments rely heavily on LLM-based judges or proxy evaluators, which introduces uncertainty about whether the conformal guarantees correspond to the intended real-world property. The paper does not analyze judge reliability or robustness to alternative evaluators.
- The empirical section is broad but somewhat shallow per task. Only the QA experiment includes a direct comparison to prior conformal methods; other experiments mainly demonstrate feasibility rather than strong performance improvements.
- The limitations and societal impact discussion is minimal, especially given the use of LLM judges, agentic web interaction, and memorization analysis in image generation.
- The presentation is not very easy to follow; the authors could alternatively state the main idea more clearly, with a central definition / theorem and a motivating example, and shift details discussion in the Appendix.

---

> ### Author Rebuttal · Authors · 2026-03-27
>
> Thank you for reviewing our work. We are glad you found that our “theoretical contribution is meaningful” and that the “topic is timely and relevant”. Below we address the issues you raised in your review; please let us know if you have any lingering concerns. We believe the discussion below addresses your concerns, if you agree, we ask that you please consider raising your score. We will clarify the points below in our camera-ready version if our paper is accepted.
>
> > On checking for $\gamma$-sensibility:
>
> Properties (b2) and (c) are always ensured by construction. Property (a), exchangeability, is usually not formally checked, but rather assumed due to knowledge of the particular task at hand. Although this is reasonable, distribution shifts between calibration and test data can cause conformal guarantees to fail, and these shifts might not be empirically detectable when they are small. Note that this is true not only of Conf-Gen, but also of CP and CRC. Lastly, the monotonicity property, (b1), is more task-dependent. In some settings, almost-sure monotonicity will hold by construction and the condition will trivially hold: **note that this is actually the case for all of our experiments except those for generation of non-memorized images**. When monotonicity does not hold almost-surely, then much like exchangeability, property (b1) cannot be formally verified to hold in practice, but nonetheless we can often reasonably assume it due to task knowledge. In the generation of non-memorized images task, property (b1) can simply be understood as saying that, in (conditional) expectation, modifying more tokens of the prompt of a memorized image will produce a less memorized image. We believe that **in this case, this property is reasonable enough to not need additional verification**. Please see also the first paragraph in Appendix C.1, where we discuss property (b1) in detail.
>
> > On the use and sensitivity with respect to LLM judges:
>
> **We believe that this concern arises due to a misunderstanding of our experiments**: note that the only experiment where the admissibility function depends on an LLM judge is the experiment in section 6.4 (agentic web-based tasks). In the question answering task, the ground-truth answers are provided as part of the dataset and there is no LLM judge whatsoever, and the conversational chatbot task also has well-defined admissibility labels. The LLM judge in the conversational chatbot is only used to construct the score function, so **the conformal guarantee does not depend on the judge** as it guarantees actual admissibility, not admissibility as ruled by an LLM judge. Please see Appendix E for all these experimental details.
>
> Lastly, we also point out that for the web agent experiments, we followed the exact setup of He et al. (2024), who provided the task dataset along with the LLM judge. He et al. (2024) made sure the considered tasks are such that the judge can accurately assess success, i.e. **we are using an already tested judge**. For this reason, we do not believe ablating the judge is necessary. We also highlight that our method is orthogonal to whether the admissibility is defined through an LLM judge or by human evaluators; when a judge is used, the conformal guarantee simply ensures admissibility as assessed by the judge. We do not see this as a downside, it is simply up to practitioners to not misunderstand the guarantee and to use an appropriate admissibility function.
>
> > On additional comparisons to baseline methods:
>
> All experiments, except those about question answering, are performed on new tasks for which **no baseline method exists**. We believe one of the key strengths about our work is extending the scope of conformal methods to these new tasks, but this does mean that there are no baselines for us to compare against. We also highlight that Appendix E includes additional experiments for both the question answering and the random forest tasks.
>
> > Clarification on the image experiments:
>
> Please see point 5 in the rebuttal to reviewer **dCAd** for an explanation of our labelling scheme. Could you please clarify which agreement metric you would like to see? Note that admissibility is defined by averaging annotator responses.
>
> > On ethics concerns and societal impact discussion:
>
> We commit to updating the current boilerplate social impact section in the camera-ready. We omit the updated section from this response due to the character limit, but will happily include it in a subsequent response if requested. Note also that our institution has no IRB requirement; see the discussion in lines 968-975.
>
> > On the presentation being hard to follow:
>
> Could you please clarify what you would like us to update? We believe Figure 1 provides a clear intuitive explanation of the method, Section 3 outlines the theory alongside a running example, Section 4 provides additional examples and discussions; and many details are already discussed in the Appendix.

---

> > ### Author Rebuttal · Reviewer_fyDF · 2026-04-03
> >
> > I'd like to thank the authors for clarifying my questions regarding the methodology, assumptions and evaluations. Some remaining concerns would be if the authors could go deeper into some of the experiments, e.g. concrete results, the amount of coverage achievable, pitfalls etc. Right now it feels more like a "quick confirmation of the theory", rather than going in depth. Besides that, I'll read also the remarks by the other reviewers and cross-check.

---

> > > ### Author Response · Authors · 2026-04-06
> > >
> > > Thank you for your continued engagement! We address your follow-up questions below, and will include these discussions in the camera-ready. Once again, please let us know if you have any lingering concerns, and we kindly ask that you consider raising your score if our points below address your concerns.
> > >
> > >
> > > > Achievable coverage and pitfalls
> > >
> > > Theoretically, any coverage $\gamma$ is achievable, but there is no guarantee that the coverage is achievable in a useful manner. This is a standard pitfall of conformal methods; for example, in conformal prediction, the outputted set could contain all possible labels, which achieves maximal coverage, but does so in a useless way. Similarly, Conf-Gen guarantees admissibility but not that it is achieved in a useful way. For example, in the task of generating non-memorized images, the guarantee could be achieved by always producing “medium” images (recall that Appendix E.2 defines “good”, “medium”, and “bad” images), which would be useless.
> > >
> > > > Deeper discussion of experiments and why we go beyond just a “quick confirmation of the theory”
> > >
> > > Precisely because of the above pitfall, all the figures in section 6 follow the same pattern: the left plots aim to empirically verify that the conformal guarantee holds, while the **right plots aim to show the method is actually useful in practice**. The right plot in: Figure 2 shows that, up to $\gamma \approx 0.84$, we can satisfy the conformal guarantee while producing small sets of responses; Figure 3 shows the percentage of “medium” images is well below $\gamma$, meaning that a a sizeable fraction of the generated images are “good”; Figure 4 shows the method does not automatically default to asking the maximal number of clarifications until $\gamma \approx 0.9$; Figure 5 shows that, below $\gamma \approx 0.66$, our method requires few tries to achieve its admissibility target.
> > >
> > > The practical usefulness of conformal methods is typically dictated by how good the score function/selection function is: the more predictive of admissibility the score function, the more useful we should expect the selected output to be. In this light, all the right plots mentioned above show that the score and selection functions we chose achieve their conformal guarantee in a non-vacuous way, and thus that **we go beyond just a “quick confirmation of the theory”** (in some cases, the usefulness holds for a specific range of values of $\gamma$, as is standard for conformal methods). This being said, we do expect future work to find even better score and selection functions that will result in even more useful Conf-Gen instances.

---

### Official Review · Reviewer_dCAd · 2026-03-13

**Soundness:** 3
**Presentation:** 2
**Significance:** 3
**Originality:** 3
**Overall Recommendation:** 4
**Confidence:** 3

**Summary:**

This paper presents a generalisation of the Conformal Risk Control (CRC) framework, Conf-Gen, to generative modelling. The paper derives the appropriate theorems for guaranteeing conformality through their construction of the selection functions and admissibility functions. The paper takes a couple of examples from text-QA, image generation, agentic decision making, etc., and shows that Conf-Gen can be used to provide a solid base for UQ in generative modelling.

**Compliance With Llm Reviewing Policy:**

Affirmed.

**Final Justification:**

The authors have broadly clarified my concerns. I believe that this is a technically sound paper with broad applicability. I have updated my score to Weak Accept, though it is dependent on the authors significantly improving their presentation of results.

However, this is not my main domain of expertise. Therefore, I cannot recommend with high confidence.

**Key Questions For Authors:**

Kindly check the weaknesses. The questions have been asked there for continuity.

**Limitations:**

Yes

**Strengths And Weaknesses:**

## Strengths:
1. The paper attempts to solve an important problem for LLM validation and uncertainty quantification.
2. The authors propose a variety of test cases with well-thought-out metrics and clever experimental design to show the working of their Conf-Gen framework in practical settings.
3. Three out of five tasks clearly show the conformal guarantees. (Remaining $2$ tasks discussed below).
4. In the Chatbot (Experiment 6.3) and Web-Agent tasks (6.4), Conf-Gen provides a basis for deciding when a model should stop and ask for clarification versus when it has enough information to act. This is a significant step toward the practical deployment of autonomous agents.
5. Remarkly self-contained paper; clarity of writing is commendable. (see weaknesses below)


## Weaknesses:
1. The discussion about the results in the main paper is weak. Yes, authors verify conformal guarantee and efficiency plots. However, there are non-trivial trends in the plots that need explanations.
2. There is significant notational overloading and mishaps:
(LC: left column; RC: right column)
    - Line 20-30; LC: There are mentions of both $S$ and $S'$ (similarly in Table 1). I am not sure why that should be the case. Please clarify.
    - Eq. (6), (7): Please use another symbol, instead of $\gamma$, to denote the second input to function $\mathcal V$. Overloading notations like this compromises readability.
3. The lower bound is an important aspect of interpreting the results; it has not been mentioned in the main paper, only in the appendix.
4. Regarding Open Domain Question Answering results:
    - Why does the Mean Test Admissibility (MTA) fall below the lower bound? I understand it as a strict requirement as per the construction, but there are multiple $\gamma$ values for MTA dips below a lower bound line.
    - The abrupt jump in sequence for high values of $\gamma$ is interesting; it suggests the model hits a limit where it can no longer be "both safe and concise". Is there a utility for such an answer by the model? Is this a property of the model of choice, LLaMa-13B or something more general, say the selection function or admissibility choices?
    - The chosen metric here is ‘at least one correct response be present in the output’, which is a weak admissibility criterion. It would be insightful if the authors could provide stronger metrics (like Exact Match or F1-score between output and GT, etc.).
    - I am curious to know if the framework is robust for smaller models (e.g., LLaMA-7B). Smaller models have higher hallucination rates; those would be a sharper test case for conformal prediction.
5. Regarding the non-memorised image generation task:
    - Same problem in the conformal guarantee plot as above. Please explain why MTA goes below the lower bound.
    - The authors define $\tau$ in Eq. (11); in case the $\tau$ index is the smallest value of $t$ for which $\max (S_i) > \lambda$. The classifier-free guidance term is the score metric. The sequence is ordered such that $X_1$ is the original prompt (high memorisation score) and $X_T$ is the most modified (therefore, least likely to be memorised). Given that $X_1$ has the highest CFG norm, the condition $\max(S_1, \dots, S_t) > \lambda$ will likely be satisfied immediately at $t=1$. This would cause the framework to select the memorised image rather than the unique ones. Please clarify the score construction.
    - The authors define admissibility as the percentage of human reviewers with responses "Good" and "Medium" (which they also take to mean “Too Different”) for the output of a given image. The framework guarantees non-memorisation; however, it allows images that are completely unrelated to the original prompt. How would the result for the conformal guarantee change if you only consider “Good” images for the admissibility?
6. Regarding Chatbot Task:
    - Similar issue as image task, please clarify the score construction.
      The chatbot starts with a very ambiguous question ($X_1$), $S_1$ will be high. The chatbot will answer if $S_1 > \lambda$, which is most likely after the first step. The score metric is ambiguous as the descriptions in the main text and appendix seem contradictory.


## Reason for the score:
This paper pitches a framework to solve the key problem of uncertainty quantification in generative modelling. The lower bound violation for conformal guarantees in two of the tasks, as well as the lack of clarity in the score function for some experiments, makes this contribution fall short of the mark.

---

> ### Author Rebuttal · Authors · 2026-03-27
>
> Thank you for your thorough review of our work. We are glad you found we used “well-thought-out metrics and clever experimental design” and that our writing is “commendable”. Below we address the issues you raised in your review; please let us know if you have any lingering concerns. We believe the discussion below addresses your concerns, if you agree, we ask that you please consider raising your score. We will clarify the points below in our camera-ready version if our paper is accepted.
>
> > (4 & 5) You mentioned in your review that only “three out of five tasks clearly show the conformal guarantees”:
>
> We kindly note that these comments are likely due to a misunderstanding of the conformal guarantees themselves. All expectations in conformal guarantees are taken with respect to all random quantities inside the expectation, including the calibration data and test point.  This expectation must be approximated with finite samples, and thus **small dips below the target test admissibility $\gamma$ in our plots do not provide evidence against the conformal guarantee**. This is not unique to Conf-Gen, CP and CRC exhibit the same behaviour. In short, **nothing in our experiments suggests any of our conformal guarantees do not hold**: only consistent or large violations would suggest otherwise.
>
> > (5 & 6) You asked about the score functions in 2 tasks.
>
> For the non-memorized image generation task, this is indeed an issue with our writing, thanks for catching it! The CFG norm is actually the negative score, not the score itself. This means equation 40 in the appendix should define $S_t’$, not $S_t$ (where $S_t’=-S_t$). This way, the selection function returns the first image with a **small** enough CFG norm, and does not typically default to the first image (which likely has the largest CFG norm).
>
> For the chatbot task, note that the **score is large when the question is clear**, not when it is ambiguous. We thus expect $S_t$ to increase as $t$ increases, and in turn the selection function will not tend to stop at the first timestep.
>
> > 1:
>
> Could you please clarify what trends you would like us to discuss further?
>
> > 2:
>
> Note that $S$ corresponds to scores for which large values tend to be associated with high admissibility values, and $S’$ to scores for which large values tend to be associated with low admissibility values. We tried to convey this in footnote 1, but will clarify further. We agree about eq 7 and will change it.
>
> > 3:
>
> We agree, but we included it in the appendix due to space constraints.
>
> > 4:
>
> The change around $\gamma \approx 0.84$ corresponds to when outputting every single response is insufficient to ensure at least one response is correct with probability $\gamma$. Indeed, at this point the model cannot be both correct and concise, and we agree that setting such a large value of $\gamma$ would not be useful in practice for this model: this can be understood as “abstaining” from making a non-vacuous guarantee (this is analogous to the prediction set being $\mathcal{Y}$ in CP). We should always expect such a jump for large enough $\gamma$, but the exact value of $\gamma$ at which the jump happens depends on the underlying LLM (if the LLM is better at producing correct answers, the jump will happen later), as well as the score and admissibility functions (the more strongly the score predicts admissibility, the later the jump will happen). Note that this is expected behaviour as there is no free lunch: if we want a very strong conformal guarantee (large $\gamma$), we might need to use very conservative sets.
>
> As for our choices of admissibility and LLaMa model: we chose these to replicate our baseline’s setup and use their existing data. In preliminary experiments we observed the same trends for LLaMA-7B, although with worse performance, both for our method and the baseline. We will add these results to the camera-ready version. Note also that changing admissibility from “at least one correct answer”, e.g. to the F1 score, could violate the monotonicity requirement.
>
> > 5:
>
> Note that **we cannot define admissibility to only include “good” images because the monotonicity requirement would be violated**: as $\lambda$ increases, the selected image needs to become more admissible (in conditional expectation). As $\lambda$ increases and we select an image corresponding to a progressively more different prompt than $X_1$, we should not expect the corresponding image to always be more “good” because it could potentially become more “medium”.

---

> > ### Author Rebuttal · Reviewer_dCAd · 2026-04-04
> >
> > Thanks for the your clarifications in the rebuttal. My W1 was supposed to be preamble, not a separate weakness. I have some more questions.
> > 1. "Small dips below the target test admissibility $\gamma$ ...": Can the allowed dip be quantified using  error bars on the MTA?
> > 2. You mention some choices could violate monotonicity violations (F1 score or exact match), I would like to know how.
> > 3. About only selection "good" images and violation of monotonicity: I agree with your reasoning. I think the definition of success here should be made clear, then: not memorised; which might mean that you will get images that have completely lost the original prompt's intent.
> >    Let me rephrase my earlier question W5.3: why is this a good test for generating good quality images? What your model guarantees is that the image will not be a duplicate, but it can be significantly different from what is mentioned in the prompt.
> > 4. (minor) Reporting the 'Good' rate independently (i.e., Admissibility excluding 'Medium/Unrelated' cases) would be highly instructive. While I acknowledge that the monotonicity requirements for the formal CRC guarantee necessitate the inclusion of the 'Medium' category, visualising the 'Good-only' rate would clarify how generative utility decays as the safety threshold $\gamma$ increases. This is a heuristic to help develop intuition for the framework's practical limits; I leave the final call to the authors.

---

> > > ### Author Response · Authors · 2026-04-06
> > >
> > > Thank you for your continued engagement! We address your follow-up questions below. Once again, please let us know if you have any lingering concerns, and we kindly ask that you consider raising your score if our points below address your concerns.
> > >
> > > > 1
> > >
> > > Although we can compute the standard deviation of the admissibility of the selected output on the test data, we cannot obtain different realizations of the calibration dataset itself, so plotting truly meaningful error bars is not directly possible. This is why papers using conformal methods do not typically report error bars. That being said, in the case of conformal prediction where the admissibility is binary, the probability of deviating by at least $\Delta$ from the bound for a given calibration dataset is at most $e^{-2n\Delta^2}$, where $n$ is the size of the calibration dataset (see Theorem 4.1 in [1]). This result can be used to flag highly unusual deviations from the bound: **we observed none in our experiments**.
> > >
> > > > 2
> > >
> > > Note that we are using exact match as the admissibility for the question answering experiments; this choice does not violate monotonicity. The F1 score is a ratio computed from a given selected sequence. When the admissibility is given by a ratio, it can be difficult to ensure the admissibility of selected sequences remains non-decreasing on $\lambda$ unless the numerator is non-decreasing **and** the denominator is non-increasing. The F1 score, $2TP / (2TP + FP + FN)$, need not satisfy this property: although the number of true positives (TP) is non-decreasing on $\lambda$, the number of incorrect answers in the outputted sequence is also non-decreasing with $\lambda$, so the denominator is non-decreasing instead of non-increasing.
> > >
> > > > 3 & 4
> > >
> > > Indeed, what our method ensures is the generated image will not be a duplicate. The setting we had in mind here is where there’s a risk of copyright infringement for producing a duplicated image, so that the model provider might prefer producing an unrelated image rather than a duplicate one. This is the setting in [2], except they provide no conformal guarantees for their method to alleviate memorization. In this case, the conformal guarantee allows the model provider to bound their risk. Avoiding copyright infringement can be highly valuable to model providers, this is precisely why we believe this task highlights the relevance of Conf-Gen.
> > >
> > > In this view, we can understand “good” as a proxy for those images which retain the original theme without being at risk of copyright infringement, and “medium” as a proxy for images which are also not a copyright infringement risk, but at the cost of losing the original theme. Of course, in practice, the usefulness of not generating copyrighted images depends on the method generating enough “good” samples rather than defaulting to producing “medium” ones: this is precisely what the right plot in Figure 3 is trying to illustrate: the fact that the percentage of generated “medium” images is well below $\gamma$ means that a meaningful fraction of the generated images are “good” images. We will explicitly report the percentage of “good” images in the camera-ready, but please note that we already report enough information to recover this value: the left plot of Figure 3 shows the percentage of [“good” or “medium”] images, and the right plot the percentage of “medium” images, so their difference provides the percentage of “good” images.
> > >
> > > [1] Theoretical Foundations of Conformal Prediction, Angelopoulos et al., arXiv preprint
> > >
> > > [2] A Geometric Framework for Understanding Memorization in Generative Models, Ross et al., ICLR

---

### Decision · Program_Chairs · 2026-04-30

**Decision:**

Accept (regular)

**Comment:**

The paper introduces Conf-Gen, a framework that extends conformal risk control (CRC) to generative models by defining score functions, admissibility functions, and a calibration procedure to provide guarantees on generated outputs . Reviewers agree that this is an important problem. One reviewer gives a strong positive evaluation, highlighting the soundness, originality, and potential impact of the work, as well as the extension of CRC with weaker assumptions (e.g., γ-sensibility).

At the same time, some concerns remain in the reviews. In particular, reviewers question how different the method is from existing CRC formulations and whether the theoretical contribution is fully clear. They also point to limitations in the empirical evaluation, which is broad across tasks (e.g., QA, image generation, agents) but shallow, with limited comparisons. Additional concerns include how to verify the assumptions (such as γ-sensibility and monotonicity) in practice, and the use of proxy evaluators in some experiments. The rebuttal clarifies some of these points, but not all concerns are fully resolved (see the reviews for details).

Please ensure that the reviewers' suggestions are incorporated into the paper in the final submission.